# Online Frank-Wolfe with Arbitrary Delays

**Yuanyu Wan**[1,2], **Wei-Wei Tu**[3,4], **Lijun Zhang**[4,*]

[1]School of Software Technology, Zhejiang University, Ningbo, China
[2]Zhejiang University-China Southern Power Grid Joint Research Centre on AI, Hangzhou, China
[3]4Paradigm Inc., Beijing, China
[4]National Key Laboratory for Novel Software Technology, Nanjing University, Nanjing, China
wanyy@zju.edu.cn, tuweiwei@4paradigm.com, zhanglj@lamda.nju.edu.cn

## Abstract

The online Frank-Wolfe (OFW) method has gained much popularity for online convex optimization due to its projection-free property. Previous studies show that OFW can attain an $O(T^{3/4})$ regret bound for convex losses and an $O(T^{2/3})$ regret bound for strongly convex losses. However, they assume that each gradient queried by OFW is revealed immediately, which may not hold in practice and limits the application of OFW. To address this limitation, we propose a delayed variant of OFW, which allows gradients to be delayed by arbitrary rounds. The main idea is to perform an update similar to OFW after receiving any delayed gradient, and play the latest decision for each round. Despite its simplicity, we prove that our delayed variant of OFW is able to achieve an $O(T^{3/4} + dT^{1/4})$ regret bound for convex losses and an $O(T^{2/3} + d\log T)$ regret bound for strongly convex losses, where $d$ is the maximum delay. This is quite surprising since under a relatively large amount of delay (e.g., $d = O(\sqrt{T})$ for convex losses and $d = O(T^{2/3}/\log T)$ for strongly convex losses), the delayed variant of OFW enjoys the same regret bound as that of the original OFW.

## 1 Introduction

Online convex optimization (OCO) has become a leading paradigm for online learning due to its capability to model various problems from diverse domains such as online routing, online collaborative filtering, and online advertisement [Hazan, 2016]. In general, it is formulated as a structured repeated game between a player and an adversary. In each round $t$, the player first chooses a decision $\mathbf{x}_t$ from a convex decision set $\mathcal{K} \subseteq \mathbb{R}^n$, where $n$ is the dimensionality. Then, the adversary selects a convex function $f_t(\mathbf{x}) : \mathcal{K} \mapsto \mathbb{R}$, and the player suffers a loss $f_t(\mathbf{x}_t)$. The player aims to choose decisions such that the regret $R(T) = \sum_{t=1}^{T} f_t(\mathbf{x}_t) - \min_{\mathbf{x} \in \mathcal{K}} \sum_{t=1}^{T} f_t(\mathbf{x})$ is sublinear in the number of total rounds $T$. Online gradient descent (OGD) is a standard method for OCO, which enjoys an $O(\sqrt{T})$ regret bound for convex losses [Zinkevich, 2003] and an $O(\log T)$ regret bound for strongly convex losses [Hazan et al., 2007]. However, it needs to compute a projection onto the decision set to ensure the feasibility of each decision, which is computationally expensive for complex decision sets [Hazan and Kale, 2012].

To tackle this computational issue, Hazan and Kale [2012] propose the online Frank-Wolfe (OFW) method, which has become one of the most commonly used algorithms for OCO over complex decision sets. The main advantage of OFW is its projection-free property: instead of performing the projection operation, it utilizes a linear optimization step to select a feasible decision, which could be much more efficient. For example, in the problem of online collaborative filtering, the decision set

---

*Lijun Zhang is the corresponding author.

36th Conference on Neural Information Processing Systems (NeurIPS 2022).

consists of matrices with a bounded trace norm, and the linear optimization step is at least an order of magnitude faster than the projection operation [Hazan and Kale, 2012]. Moreover, it has been shown that OFW achieves an $O(T^{3/4})$ regret bound for convex losses [Hazan and Kale, 2012, Hazan, 2016] and an $O(T^{2/3})$ regret bound for strongly convex losses [Wan and Zhang, 2021, Garber and Kretzu, 2021], which are the best known regret bounds of projection-free methods without further assumptions.

However, OFW requires that the gradient $\nabla f_t(\mathbf{x}_t)$ is revealed immediately after making the decision $\mathbf{x}_t$, which is not necessarily satisfied in reality. For example, in the previously mentioned online collaborative filtering [Hazan and Kale, 2012], the decision is a prediction of a user-item rating matrix, and the corresponding gradient depends on the true rating of a user on a item, which may not be decided by the user immediately. Therefore, it is natural to consider a more practical setting, where the gradient $\nabla f_t(\mathbf{x}_t)$ arrives at the end of round $t + d_t - 1$, and $d_t \geq 1$ denotes an arbitrary delay. To handle this setting, one potential way is to combine OFW with an existing black-box technique for converting any traditional OCO algorithm into this delayed setting [Joulani et al., 2013]. To be precise, this black-box technique is to pool independent instances of OFW, each of which acts as a learner in the non-delayed setting over a subsequence of rounds. In each round, a single instance will be taken out from the pool, which makes a decision and then waits for its feedback before rejoining the pool. If the pool is empty, a new instance of OFW will be added to it. Moreover, according to Joulani et al. [2013], their black-box technique is able to attain a regret bound of $dR(T/d)$ by combining with a traditional OCO algorithm with $R(T)$ regret, where $d$ is the maximum delay. As a result, combining it with OFW will attain an $O(d^{1/4}T^{3/4})$ regret bound for convex losses and an $O(d^{1/3}T^{2/3})$ regret bound for strongly convex losses, which magnify the regret bounds of OFW in the non-delayed setting by a coefficient depending the delay. Thus, it is natural to ask whether the effect of delay can be further reduced.

In this paper, we give an affirmative answer by developing a simple method called delayed OFW, which is robust to a relatively large amount of delay for both convex and strongly convex losses. Different from the black-box technique that needs to maintain multiple instances of OFW [Joulani et al., 2013], our delayed OFW is a natural extension of OFW in the delayed setting, which updates the decision similar to OFW after receiving any delayed gradient, and plays the latest decision for each round. Our theoretical contributions are summarized as follows.

- First, we prove that our delayed OFW attains an $O(T^{3/4} + dT^{1/4})$ regret bound for convex losses, where $d$ is the maximum delay, which matches the $O(T^{3/4})$ regret bound in the non-delayed setting as long as $d$ does not exceed $O(\sqrt{T})$.

- Second, we prove that our delayed OFW attains an $O(T^{2/3} + d\log T)$ regret bound for strongly convex losses, which matches the $O(T^{2/3})$ regret bound in the non-delayed setting as long as $d$ does not exceed $O(T^{2/3}/\log T)$.

Therefore, our regret bounds are strictly better than those achieved by combining the black-box technique [Joulani et al., 2013] with OFW, when the term involving $d$ in them is not dominant. Furthermore, simulation experiments are conducted to verify the performance of our delayed OFW.

## 2  Related work

In this section, we briefly review related work on projection-free algorithms for OCO, and OCO under delayed feedback.

### 2.1  Projection-free algorithms for OCO

The OFW method [Hazan and Kale, 2012, Hazan, 2016] is the first projection-free algorithm for OCO, which is an online extension of the classical Frank-Wolfe method [Frank and Wolfe, 1956, Jaggi, 2013]. For convex losses, OFW first chooses an arbitrary $\mathbf{x}_1 \in \mathcal{K}$, and then iteratively updates its decision by the following linear optimization step

$$\mathbf{v}_t \in \operatorname*{argmin}_{\mathbf{x} \in \mathcal{K}} \langle \nabla F_t(\mathbf{x}_t), \mathbf{x} \rangle, \ \mathbf{x}_{t+1} = \mathbf{x}_t + \sigma_t(\mathbf{v}_t - \mathbf{x}_t) \tag{1}$$

where $F_t(\mathbf{x})$ is a surrogate loss function defined as

$$F_t(\mathbf{x}) = \eta \sum_{i=1}^{t} \langle \nabla f_i(\mathbf{x}_i), \mathbf{x} \rangle + \|\mathbf{x} - \mathbf{x}_1\|_2^2 \tag{2}$$

and $\eta, \sigma_t$ are two parameters. By setting parameters appropriately, it can attain an $O(T^{3/4})$ regret bound for convex losses.

If losses are convex and smooth, Hazan and Minasyan [2020] propose a randomized projection-free method, which is based on a classical OCO method called follow the perturbed leader [Kalai and Vempala, 2005], and achieve a regret bound of $O(T^{2/3})$. Recently, Wan and Zhang [2021] prove that OFW can achieve an $O(T^{2/3})$ regret bound for strongly convex losses. Specifically, to utilize the strong convexity of losses, they redefine $F_t(\mathbf{x})$ in (2) to

$$F_t(\mathbf{x}) = \sum_{i=1}^{t} \left( \langle \nabla f_i(\mathbf{x}_i), \mathbf{x} \rangle + \frac{\beta}{2} \|\mathbf{x} - \mathbf{x}_i\|_2^2 \right) \tag{3}$$

where $\beta$ is the modulus of the strong convexity. The same regret bound is concurrently established by Garber and Kretzu [2021] in a similar way.

Moreover, projection-free OCO algorithms, which are able to achieve an $O(\sqrt{T})$ regret bound for convex losses and an $O(\log T)$ regret bound for strongly convex losses, have been proposed for polyhedral sets [Garber and Hazan, 2016], smooth sets [Levy and Krause, 2019], and special sets that are accessible through a membership oracle [Mhammedi, 2022] or a separation oracle [Garber and Kretzu, 2022], respectively. Although regret bounds of these algorithms are better than those of OFW, they can only be efficiently implemented in some special cases, and thus are still not able to replace OFW. In addition to these algorithms, Wan and Zhang [2021] show that by setting $\sigma_t$ with a line search technique, OFW can adaptively utilize the strong convexity of decision sets to attain an $O(T^{2/3})$ regret bound for convex losses and an $O(\sqrt{T})$ regret bound for strongly convex losses. Besides the standard setting, projection-free algorithms for other OCO scenarios such as the distributed setting [Zhang et al., 2017, Wan et al., 2020, 2022b] and the bandit setting [Chen et al., 2019, Garber and Kretzu, 2020, 2021] have also been proposed.

However, despite this great flourish of research on projection-free OCO algorithms, the practical problem of delayed feedback has not been considered.

## 2.2 OCO under delayed feedback

The starting point for studies on OCO under delayed feedback is the seminal work of Weinberger and Ordentlich [2002], which focuses on a special case with a fixed delay $d'$, i.e., the feedback for each decision $\mathbf{x}_t$ is received at the end of round $t + d' - 1$. They propose a technique that can convert any traditional OCO algorithm for the non-delayed setting into this delayed setting. Specifically, their technique is to run $d'$ instances of a traditional OCO algorithm, where each instance is used every $d'$ rounds, which is feasible because the feedback for any decision $\mathbf{x}_t$ must be available for selecting the decision $\mathbf{x}_{t+d'}$. If the traditional OCO algorithm enjoys a regret bound of $R(T)$ in the non-delayed setting, they prove that this technique achieves a regret bound of $d'R(T/d')$. As a result, by combining with OGD, the regret bound of this technique is $O(\sqrt{d'T})$ for convex losses and $O(d' \log T)$ for strongly convex losses. However, it needs to run $d'$ instances of OGD, which requires more storage and computational resources than the original OGD. To address this problem, Langford et al. [2009] study the same delayed setting, and show that simply performing each gradient descent step in the original OGD with a delayed gradient can also achieve the $O(\sqrt{d'T})$ regret bound for convex losses and the $O(d' \log T)$ regret bound for strongly convex losses.

Furthermore, Joulani et al. [2013] extend the technique in Weinberger and Ordentlich [2002] to handle a more general setting, where each feedback is delayed by arbitrary rounds. For this setting, their technique is able to attain a regret bound of $dR(T/d)$ by combining with a traditional OCO algorithm with $R(T)$ regret, where $d$ is the maximum delay. Note that although this technique can also convert OFW to the delayed setting, it can only attain an $O(d^{1/4}T^{3/4})$ regret bound for convex losses and an $O(d^{1/3}T^{2/3})$ regret bound for strongly convex losses, which cannot match regret bounds of OFW in the non-delayed setting as long as $d$ is larger than $\Omega(1)$. Additionally, similar to the technique

in Weinberger and Ordentlich [2002], it also needs to run multiple instances of a traditional OCO algorithm, which could be prohibitively resource-intensive. Many studies [Quanrud and Khashabi, 2015, Joulani et al., 2016, Li et al., 2019, Flaspohler et al., 2021, Wan et al., 2022a] have proposed delayed OCO algorithms, which only require the same storage and computational resources as in the non-delayed setting, but do not consider projection-free algorithms.

## 3 Main results

In this section, we first introduce necessary preliminaries including the problem setting, definitions, and assumptions. Then, we present our delayed OFW and the corresponding theoretical guarantees for convex and strongly convex losses, respectively.

### 3.1 Preliminaries

We consider the problem of OCO with arbitrary delays [Joulani et al., 2013, Quanrud and Khashabi, 2015]. Similar to the standard OCO, in each round $t = 1, \ldots, T$, the player first chooses a decision $\mathbf{x}_t$ from the decision set $\mathcal{K}$, and then the adversary selects a convex function $f_t(\mathbf{x})$. However, different from the standard OCO, the gradient $\mathbf{g}_t = \nabla f_t(\mathbf{x}_t)$ is revealed at the end of round $t + d_t - 1$, where $d_t \geq 1$ denotes an arbitrary delay. As a result, the player actually receives gradients $\{\mathbf{g}_k | k \in \mathcal{F}_t\}$ at the end of round $t$, where $\mathcal{F}_t = \{k | k + d_k - 1 = t\}$. Then, we recall the standard definition for strongly convex functions [Boyd and Vandenberghe, 2004].

**Definition 1** *A function $f(\mathbf{x}) : \mathcal{K} \to \mathbb{R}$ is called $\beta$-strongly convex over $\mathcal{K}$ if for all $\mathbf{x}, \mathbf{y} \in \mathcal{K}$, it holds that $f(\mathbf{y}) \geq f(\mathbf{x}) + \langle \nabla f(\mathbf{x}), \mathbf{y} - \mathbf{x} \rangle + \frac{\beta}{2} \|\mathbf{y} - \mathbf{x}\|_2^2$.*

Finally, we introduce two assumptions, which are commonly used in studies about OCO [Shalev-Shwartz, 2011, Hazan, 2016].

**Assumption 1** *The gradients of all loss functions are bounded by $G$, i.e., it holds that $\|\nabla f_t(\mathbf{x})\|_2 \leq G$ for any $\mathbf{x} \in \mathcal{K}$ and $t \in [T]$.*

**Assumption 2** *The diameter of the decision set $\mathcal{K}$ is bounded by $D$, i.e., it holds that $\|\mathbf{x} - \mathbf{y}\|_2 \leq D$ for any $\mathbf{x}, \mathbf{y} \in \mathcal{K}$.*

### 3.2 Delayed OFW for convex losses

As presented in (1) and (2), the original OFW for convex losses requires the gradient $\mathbf{g}_t$ before making the decision $\mathbf{x}_{t+1}$. However, in the problem of OCO with arbitrary delays, this requirement is not necessarily satisfied, because the player actually receives gradients $\{\mathbf{g}_k | k \in \mathcal{F}_t\}$ at each round $t$, which may not contain $\mathbf{g}_t$. To address this limitation, we propose a natural generalization of OFW to this delayed problem, which is described as follows.

The main idea is to update the decision similar to OFW for each received gradient, and play the latest decision for each round. In this way, there exist some intermediate decisions that are not really played. To facilitate presentations, we introduce an additional notation $\mathbf{y}_\tau$ to denote the $\tau$-th intermediate decision. Moreover, we denote the sum of $\tau$ received gradients by $\bar{\mathbf{g}}_\tau$. Initially, we choose an arbitrary vector $\mathbf{y}_1 \in \mathcal{K}$ and set $\tau = 1, \bar{\mathbf{g}}_0 = \mathbf{0}$. At each round $t = 1, \ldots, T$, we play the latest decision $\mathbf{x}_t = \mathbf{y}_\tau$ and query the gradient $\mathbf{g}_t = \nabla f_t(\mathbf{x}_t)$. Then, we receive delayed gradients that are queried in a set of rounds $\mathcal{F}_t$, and perform the following steps for each received gradient.

Specifically, for any $k \in \mathcal{F}_t$, inspired by (2) of the original OFW, we first compute $\bar{\mathbf{g}}_\tau = \bar{\mathbf{g}}_{\tau-1} + \mathbf{g}_k$ and define

$$F_\tau(\mathbf{y}) = \eta \langle \bar{\mathbf{g}}_\tau, \mathbf{y} \rangle + \|\mathbf{y} - \mathbf{y}_1\|_2^2.$$

Then, similar to (1) of the original OFW, we perform the following update

$$\mathbf{v}_\tau \in \operatorname*{argmin}_{\mathbf{y} \in \mathcal{K}} \langle \nabla F_\tau(\mathbf{y}_\tau), \mathbf{y} \rangle, \; \mathbf{y}_{\tau+1} = \mathbf{y}_\tau + \sigma_\tau(\mathbf{v}_\tau - \mathbf{y}_\tau)$$

where $\sigma_\tau$ is a parameter. Following Wan and Zhang [2021], it is set by a line search rule

$$\sigma_\tau = \operatorname*{argmin}_{\sigma \in [0,1]} \langle \sigma(\mathbf{v}_\tau - \mathbf{y}_\tau), \nabla F_\tau(\mathbf{y}_\tau) \rangle + \sigma^2 \|\mathbf{v}_\tau - \mathbf{y}_\tau\|_2^2. \tag{4}$$

---
**Algorithm 1** Delayed OFW for Convex Losses
---
1: **Input:** $\eta$
2: **Initialization:** choose an arbitrary vector $\mathbf{y}_1 \in \mathcal{K}$ and set $\tau = 1, \bar{\mathbf{g}}_0 = \mathbf{0}$
3: **for** $t = 1, 2, \ldots, T$ **do**
4:     Play $\mathbf{x}_t = \mathbf{y}_\tau$ and query $\mathbf{g}_t = \nabla f_t(\mathbf{x}_t)$
5:     Receive a set of delayed gradients $\{\mathbf{g}_k | k \in \mathcal{F}_t\}$
6:     **for** $k \in \mathcal{F}_t$ **do**
7:         Update $\bar{\mathbf{g}}_\tau = \bar{\mathbf{g}}_{\tau-1} + \mathbf{g}_k$ and define $F_\tau(\mathbf{y}) = \eta \langle \bar{\mathbf{g}}_\tau, \mathbf{y} \rangle + \|\mathbf{y} - \mathbf{y}_1\|_2^2$
8:         Compute $\mathbf{v}_\tau \in \operatorname{argmin}_{\mathbf{y} \in \mathcal{K}} \langle \nabla F_\tau(\mathbf{y}_\tau), \mathbf{y} \rangle$
9:         Update $\mathbf{y}_{\tau+1} = \mathbf{y}_\tau + \sigma_\tau(\mathbf{v}_\tau - \mathbf{y}_\tau)$ with $\sigma_\tau$ in (4) and set $\tau = \tau + 1$
10:     **end for**
11: **end for**
---

Finally, we update $\tau = \tau + 1$ so that $\tau$ still indexes the latest intermediate decision.

The detailed procedures are summarized in Algorithm 1, which is named as delayed OFW for convex losses. Let $d = \max\{d_1, \ldots, d_T\}$. We establish the following theorem with respect to the regret of Algorithm 1.

**Theorem 1** *For any $\mathbf{x}^* \in \mathcal{K}$, under Assumptions 1 and 2, Algorithm 1 with $\eta = \frac{D}{\sqrt{2}G(T+2)^{3/4}}$ has*

$$\sum_{t=1}^{T} f_t(\mathbf{x}_t) - \sum_{t=1}^{T} f_t(\mathbf{x}^*) = O(T^{3/4} + dT^{1/4}).$$

Theorem 1 shows that without knowing the value of $d$, our Algorithm 1 can attain an $O(T^{3/4} + dT^{1/4})$ regret bound for convex losses with arbitrary delays. This bound matches the $O(T^{3/4})$ regret bound of OFW in the non-delayed setting [Hazan, 2016], as long as $d$ does not exceed $O(\sqrt{T})$. Moreover, it is better than the $O(d^{1/4}T^{3/4})$ regret bound achieved by combining the technique of Joulani et al. [2013] and the $O(T^{3/4})$ regret bound of OFW for convex losses, as long as $d$ does not exceed $O(T^{2/3})$.

### 3.3 Delayed OFW for strongly convex losses

We proceed to handle $\beta$-strongly convex losses by slightly modifying Algorithm 1. Recall that in the standard OCO without delays, the critical idea of utilizing the strong convexity of losses is to replace the surrogate loss function in (1) by that in (3) [Wan and Zhang, 2021]. The main difference is that the regularization term in (3) is about all historical decisions, instead of only the initial decision.

Inspired by (3), we first redefine $F_\tau(\mathbf{y})$ in Algorithm 1 to

$$F_\tau(\mathbf{y}) = \langle \bar{\mathbf{g}}_\tau, \mathbf{y} \rangle + \sum_{i=1}^{\tau} \frac{\beta}{2} \|\mathbf{y} - \mathbf{y}_i\|_2^2.$$

Second, since $F_\tau(\mathbf{y})$ is modified, we adjust the line search rule to

$$\sigma_\tau = \operatorname*{argmin}_{\sigma \in [0,1]} \langle \sigma(\mathbf{v}_\tau - \mathbf{y}_\tau), \nabla F_\tau(\mathbf{y}_\tau) \rangle + \frac{\beta \tau \sigma^2}{2} \|\mathbf{v}_\tau - \mathbf{y}_\tau\|_2^2. \tag{5}$$

The detailed procedures are summarized in Algorithm 2, which is named as delayed OFW for strongly convex losses. Then, we establish the following theorem about the regret of Algorithm 2.

**Theorem 2** *Suppose all losses are $\beta$-strongly convex and Assumptions 1 and 2 hold. For any $\mathbf{x}^* \in \mathcal{K}$, Algorithm 2 has*

$$\sum_{t=1}^{T} f_t(\mathbf{x}_t) - \sum_{t=1}^{T} f_t(\mathbf{x}^*) = O(T^{2/3} + d \log T).$$

Theorem 2 shows that our Algorithm 2 can attain an $O(T^{2/3} + d \log T)$ regret bound for strongly convex losses with arbitrary delays. First, this bound is better than the $O(T^{3/4} + dT^{1/4})$ regret bound

---

**Algorithm 2** Delayed OFW for Strongly Convex Losses

---

1: **Input:** $\beta$
2: **Initialization:** choose an arbitrary vector $\mathbf{y}_1 \in \mathcal{K}$ and set $\tau = 1, \bar{\mathbf{g}}_0 = \mathbf{0}$
3: **for** $t = 1, 2, \ldots, T$ **do**
4:     Play $\mathbf{x}_t = \mathbf{y}_\tau$ and query $\mathbf{g}_t = \nabla f_t(\mathbf{x}_t)$
5:     Receive a set of delayed gradients $\{\mathbf{g}_k | k \in \mathcal{F}_t\}$
6:     **for** $k \in \mathcal{F}_t$ **do**
7:         Update $\bar{\mathbf{g}}_\tau = \bar{\mathbf{g}}_{\tau-1} + \mathbf{g}_k$ and define $F_\tau(\mathbf{y}) = \langle \bar{\mathbf{g}}_\tau, \mathbf{y} \rangle + \sum_{i=1}^{\tau} \frac{\beta}{2} \|\mathbf{y} - \mathbf{y}_i\|_2^2$
8:         Compute $\mathbf{v}_\tau \in \operatorname{argmin}_{\mathbf{y} \in \mathcal{K}} \langle \nabla F_\tau(\mathbf{y}_\tau), \mathbf{y} \rangle$
9:         Update $\mathbf{y}_{\tau+1} = \mathbf{y}_\tau + \sigma_\tau(\mathbf{v}_\tau - \mathbf{y}_\tau)$ with $\sigma_\tau$ in (5) and set $\tau = \tau + 1$
10:    **end for**
11: **end for**

---

in Theorem 1, which is established by only using the convexity condition. Second, it matches the $O(T^{2/3})$ regret bound of OFW for strongly convex losses in the non-delayed setting [Wan and Zhang, 2021], as long as $d$ does not exceed $O(T^{2/3}/\log T)$. Third, it is better than the $O(d^{1/3}T^{2/3})$ regret bound achieved by combining the technique of Joulani et al. [2013] and the $O(T^{2/3})$ regret bound of OFW for strongly convex losses, as long as $d$ does not exceed $O(T/(\log T)^{3/2})$.

## 4 Theoretical analysis

In this section, we first introduce some insights and preliminaries for our analysis, and then prove Theorems 1 and 2 by introducing some lemmas. Due to the limitation of space, the proofs for those lemmas are provided in the appendix.

### 4.1 Insights and preliminaries

Note that in the non-delayed setting, the regret bound of OFW is worse than that of OGD due to the gap between the linear optimization step and the projection operation. Thus, in the delayed setting, it is natural to conjecture that both the linear optimization step and the delay will affect the regret of our delayed OFW. To keep the same regret bound as the original OFW, our key insight is to prove that the combined effect of the linear optimization step and the delay can be additive, instead of being multiplicative, and the effect of the delay can be weaker than that of the linear optimization step for a relatively large amount of delay.

Moreover, there are some necessary preliminaries for our analysis. First, because of the effect of delays, there may exist some gradients that arrive after round $T$. Although our Algorithms 1 and 2 do not need to use these gradients, they are useful for the analysis. As a result, in our analysis, we further set $\mathbf{x}_t = \mathbf{y}_\tau$ and perform steps 5 to 10 of Algorithms 1 and 2 for any $t = T+1, \ldots, T+d-1$. In this way, all gradients $\mathbf{g}_1, \mathbf{g}_2, \ldots, \mathbf{g}_T$ queried by our algorithms are utilized, which produces decisions $\mathbf{y}_1, \mathbf{y}_2, \ldots, \mathbf{y}_{T+1}$.

Then, let $\tau_t = 1 + \sum_{i=1}^{t-1} |\mathcal{F}_i|$ for any $t \in [T+d]$. It is not hard to verify that our Algorithms 1 and 2 ensure that

$$\mathbf{x}_t = \mathbf{y}_{\tau_t} \tag{6}$$

for any $t \in [T+d-1]$. Next, we define

$$\mathcal{I}_t = \begin{cases} \emptyset, & \text{if } |\mathcal{F}_t| = 0, \\ \{\tau_t, \tau_t + 1, \ldots, \tau_{t+1} - 1\}, & \text{otherwise.} \end{cases} \tag{7}$$

Let $s = \min\{t | t \in [T+d-1], |\mathcal{F}_t| > 0\}$. It is not hard to verify that

$$\cup_{t=s}^{T+d-1} \mathcal{I}_t = [T], \mathcal{I}_i \cap \mathcal{I}_j = \emptyset, \forall i \neq j, \tag{8}$$

$$\cup_{t=s}^{T+d-1} \mathcal{F}_t = [T], \mathcal{F}_i \cap \mathcal{F}_j = \emptyset, \forall i \neq j. \tag{9}$$

To facilitate the analysis, we further denote the time-stamp of the $\tau$-th gradient used in the update of our Algorithms 1 and 2 by $c_\tau$. To help understanding, one can imagine that our Algorithms 1 and 2 also implement $c_\tau = k$ in their step 7. If $|\mathcal{F}_t| \neq 0$, we have

$$\{c_{\tau_t}, \ldots, c_{\tau_{t+1}-1}\} = \mathcal{F}_t. \tag{10}$$

By using this notation, $F_\tau(\mathbf{y})$ defined in Algorithms 1 and 2 are respectively equivalent to

$$F_\tau(\mathbf{y}) = \eta \sum_{i=1}^{\tau} \langle \mathbf{g}_{c_i}, \mathbf{y} \rangle + \|\mathbf{y} - \mathbf{y}_1\|_2^2, \tag{11}$$

$$F_\tau(\mathbf{y}) = \sum_{i=1}^{\tau} \langle \mathbf{g}_{c_i}, \mathbf{y} \rangle + \sum_{i=1}^{\tau} \frac{\beta}{2}\|\mathbf{y} - \mathbf{y}_i\|_2^2. \tag{12}$$

### 4.2 Proof of Theorem 1

Let $t' = t + d_t - 1$ for any $t \in [T]$. According to the convexity of $f_t(\mathbf{x})$, we have

$$f_t(\mathbf{x}_t) - f_t(\mathbf{x}^*) \leq \langle \mathbf{g}_t, \mathbf{x}_t - \mathbf{x}^* \rangle = \langle \mathbf{g}_t, \mathbf{x}_{t'} - \mathbf{x}^* \rangle + \langle \mathbf{g}_t, \mathbf{x}_t - \mathbf{x}_{t'} \rangle$$
$$\leq \langle \mathbf{g}_t, \mathbf{x}_{t'} - \mathbf{x}^* \rangle + G\|\mathbf{x}_t - \mathbf{x}_{t'}\|_2 = \langle \mathbf{g}_t, \mathbf{y}_{\tau_t} - \mathbf{x}^* \rangle + G\|\mathbf{y}_{\tau_t} - \mathbf{y}_{\tau_{t'}}\|_2$$

where the last equality is due to (6).

Let $A = \sum_{t=1}^{T} G\|\mathbf{y}_{\tau_t} - \mathbf{y}_{\tau_{t'}}\|_2$. By summing over $t = 1, \ldots, T$, we have

$$\sum_{t=1}^{T} f_t(\mathbf{x}_t) - \sum_{t=1}^{T} f_t(\mathbf{x}^*) \leq \sum_{t=1}^{T} \langle \mathbf{g}_t, \mathbf{x}_t - \mathbf{x}^* \rangle \leq \sum_{t=1}^{T} \langle \mathbf{g}_t, \mathbf{y}_{\tau_{t'}} - \mathbf{x}^* \rangle + A. \tag{13}$$

Then, we bound the first term in the right side of (13) as follows

$$\sum_{t=1}^{T} \langle \mathbf{g}_t, \mathbf{y}_{\tau_{t'}} - \mathbf{x}^* \rangle = \sum_{t=s}^{T+d-1} \sum_{i \in \mathcal{F}_t} \langle \mathbf{g}_i, \mathbf{y}_{\tau_{i+d_i-1}} - \mathbf{x}^* \rangle$$

$$= \sum_{t=s}^{T+d-1} \sum_{i \in \mathcal{F}_t} \langle \mathbf{g}_i, \mathbf{y}_{\tau_t} - \mathbf{x}^* \rangle = \sum_{t=s}^{T+d-1} \sum_{i=\tau_t}^{\tau_{t+1}-1} \langle \mathbf{g}_{c_i}, \mathbf{y}_{\tau_t} - \mathbf{x}^* \rangle$$

$$= \sum_{t=s}^{T+d-1} \sum_{i=\tau_t}^{\tau_{t+1}-1} \left( \langle \mathbf{g}_{c_i}, \mathbf{y}_i - \mathbf{x}^* \rangle + \langle \mathbf{g}_{c_i}, \mathbf{y}_{\tau_t} - \mathbf{y}_i \rangle \right) \tag{14}$$

$$= \sum_{t=1}^{T} \langle \mathbf{g}_{c_t}, \mathbf{y}_t - \mathbf{x}^* \rangle + \sum_{t=s}^{T+d-1} \sum_{i=\tau_t}^{\tau_{t+1}-1} \langle \mathbf{g}_{c_i}, \mathbf{y}_{\tau_t} - \mathbf{y}_i \rangle$$

where the first equality is due to (9), the second equality is due to $i + d_i - 1 = t$ for any $i \in \mathcal{F}_t$, the third equality is due to (10), and the last equality is due to (7) and (8).

Then, let $B = \sum_{t=1}^{T} \langle \mathbf{g}_{c_t}, \mathbf{y}_t - \mathbf{x}^* \rangle$ and $C = \sum_{t=s}^{T+d-1} \sum_{i=\tau_t}^{\tau_{t+1}-1} G\|\mathbf{y}_{\tau_t} - \mathbf{y}_i\|_2$. By combining (13), (14), and $\langle \mathbf{g}_{c_i}, \mathbf{y}_{\tau_t} - \mathbf{y}_i \rangle \leq G\|\mathbf{y}_{\tau_t} - \mathbf{y}_i\|_2$, we have

$$\sum_{t=1}^{T} f_t(\mathbf{x}_t) - \sum_{t=1}^{T} f_t(\mathbf{x}^*) \leq A + B + C. \tag{15}$$

Next, we proceed to bound terms $A$, $B$, and $C$. Specifically, we first establish the following bound for the sum of terms $A$ and $C$ by carefully analyzing the distance $\|\mathbf{y}_{\tau_t} - \mathbf{y}_{\tau_{t'}}\|_2$ in the term $A$ and the distance $\|\mathbf{y}_{\tau_t} - \mathbf{y}_i\|_2$ in the term $C$.

**Lemma 1** *Let $\mathbf{y}_t^* = \operatorname{argmin}_{\mathbf{y} \in \mathcal{K}} F_{t-1}(\mathbf{y})$ for any $t \in [T+1]$, where $F_t(\mathbf{y})$ is defined in (11). Suppose Assumption 1 and 2 hold, and there exist some constants $\gamma > 0$ and $0 < \alpha \leq 1$ such that $F_{t-1}(\mathbf{y}_t) - F_{t-1}(\mathbf{y}_t^*) \leq \gamma(t+2)^{-\alpha}$ for any $t \in [T+1]$. Algorithm 1 ensures*

$$A + C \leq 3dGD + 4Gd\sqrt{\gamma} + \frac{8G\sqrt{\gamma}}{2-\alpha}T^{1-\alpha/2} + \frac{3\eta G^2 dT}{2}$$

*where $A = \sum_{t=1}^{T} G\|\mathbf{y}_{\tau_t} - \mathbf{y}_{\tau_{t'}}\|_2$ and $C = \sum_{t=s}^{T+d-1} \sum_{i=\tau_t}^{\tau_{t+1}-1} G\|\mathbf{y}_{\tau_t} - \mathbf{y}_i\|_2$.*

Note that Lemma 1 introduces an assumption about $\mathbf{y}_t$ and $F_{t-1}(\mathbf{y})$. According to our Algorithm 1, $\mathbf{y}_t$ is actually generated by approximately minimizing $F_{t-1}(\mathbf{y})$ with a linear optimization step. Therefore, by following the analysis of the original OFW [Hazan, 2016], we show that this assumption can be satisfied with $\gamma = 8D^2$ and $\alpha = 1/2$.

**Lemma 2** *Let $\mathbf{y}_t^* = \arg\min_{\mathbf{y} \in \mathcal{K}} F_{t-1}(\mathbf{y})$ for any $t \in [T+1]$, where $F_t(\mathbf{y})$ is defined in (11). Under Assumptions 1 and 2, for any $t \in [T+1]$, Algorithm 1 with $\eta = \frac{D}{\sqrt{2}G(T+2)^{3/4}}$ has*

$$F_{t-1}(\mathbf{y}_t) - F_{t-1}(\mathbf{y}_t^*) \leq \frac{8D^2}{\sqrt{t+2}}.$$

Then, by combining $\eta = \frac{D}{\sqrt{2}G(T+2)^{3/4}}$ with Lemmas 1 and 2, we have

$$A + C \leq (3 + 8\sqrt{2})GDd + \frac{32\sqrt{2}GD}{3}T^{3/4} + \frac{3GDdT^{1/4}}{2\sqrt{2}} = O(T^{3/4} + dT^{1/4}). \quad (16)$$

Furthermore, by following the analysis of the original OFW [Hazan, 2016], we establish an upper bound for the term $B$.

**Lemma 3** *Under Assumptions 1 and 2, for any $\mathbf{x}^* \in \mathcal{K}$, Algorithm 1 with $\eta = \frac{D}{\sqrt{2}G(T+2)^{3/4}}$ ensures*

$$\sum_{t=1}^{T} \langle \mathbf{g}_{c_t}, \mathbf{y}_t - \mathbf{x}^* \rangle \leq \frac{11\sqrt{2}GD(T+2)^{3/4}}{3} + \frac{GDT^{1/4}}{\sqrt{2}}.$$

Finally, by combining (15), (16), and Lemma 3, we complete this proof.

### 4.3 Proof of Theorem 2

Since $f_t(\mathbf{x})$ is $\beta$-strongly convex, we have

$$\sum_{t=1}^{T} f_t(\mathbf{x}_t) - \sum_{t=1}^{T} f_t(\mathbf{x}^*) \leq \sum_{t=1}^{T} \langle \mathbf{g}_t, \mathbf{x}_t - \mathbf{x}^* \rangle - \sum_{t=1}^{T} \frac{\beta}{2} \|\mathbf{x}_t - \mathbf{x}^*\|_2^2. \quad (17)$$

Then, we note that the first term in the right side of (17) can be bounded by reusing (13) and (14). Specifically, we have

$$\sum_{t=1}^{T} f_t(\mathbf{x}_t) - \sum_{t=1}^{T} f_t(\mathbf{x}^*) \leq A + C + \sum_{t=1}^{T} \langle \mathbf{g}_{c_t}, \mathbf{y}_t - \mathbf{x}^* \rangle - \sum_{t=1}^{T} \frac{\beta}{2} \|\mathbf{x}_t - \mathbf{x}^*\|_2^2 \quad (18)$$

where terms $A$ and $C$ are defined in the proof of Theorem 1.

Next, we consider the last term in the right side of (18). For any $\mathbf{y}_t, \mathbf{x}_t, \mathbf{x}^* \in \mathcal{K}$, we have

$$\|\mathbf{y}_t - \mathbf{x}^*\|_2^2 = \|\mathbf{y}_t - \mathbf{x}_t\|_2^2 + \|\mathbf{x}_t - \mathbf{x}^*\|_2^2 + 2\langle \mathbf{y}_t - \mathbf{x}_t, \mathbf{x}_t - \mathbf{x}^* \rangle$$
$$\leq 3D\|\mathbf{y}_t - \mathbf{x}_t\|_2 + \|\mathbf{x}_t - \mathbf{x}^*\|_2^2$$

where the last inequality is due to $2\langle \mathbf{y}_t - \mathbf{x}_t, \mathbf{x}_t - \mathbf{x}^* \rangle \leq 2\|\mathbf{y}_t - \mathbf{x}_t\|_2 \|\mathbf{x}_t - \mathbf{x}^*\|_2$ and Assumption 2.

Let $B' = \sum_{t=1}^{T} \langle \mathbf{g}_{c_t}, \mathbf{y}_t - \mathbf{x}^* \rangle - \sum_{t=1}^{T} \frac{\beta}{2} \|\mathbf{y}_t - \mathbf{x}^*\|_2^2$ and $E = \sum_{t=1}^{T} \frac{3\beta D}{2} \|\mathbf{y}_t - \mathbf{y}_{\tau_t}\|_2$. By combining the above inequality and (6) with (18), we have

$$\sum_{t=1}^{T} [f_t(\mathbf{x}_t) - f_t(\mathbf{x}^*)] \leq A + C + B' + E. \quad (19)$$

Then, we proceed to establish upper bounds for terms $A$, $C$, and $E$ by carefully analyzing the distance $\|\mathbf{y}_{\tau_t} - \mathbf{y}_{\tau_{t'}}\|_2$ in the term $A$, the distance $\|\mathbf{y}_{\tau_t} - \mathbf{y}_i\|_2$ in the term $C$, and the distance $\|\mathbf{y}_t - \mathbf{y}_{\tau_t}\|_2$ in the term $E$.

**Lemma 4** *Let $\mathbf{y}_t^* = \arg\min_{\mathbf{y} \in \mathcal{K}} F_{t-1}(\mathbf{y})$ for any $t = 2, \ldots, T+1$, where $F_t(\mathbf{y})$ is defined in (12). Suppose Assumption 1 and 2 hold, all losses are $\beta$-strongly convex, and there exist some constants $\gamma > 0$ and $0 \leq \alpha < 1$ such that $F_{t-1}(\mathbf{y}_t) - F_{t-1}(\mathbf{y}_t^*) \leq \gamma(t-1)^\alpha$ for any $t = 2, \ldots, T+1$. Algorithm 1 ensures*

$$E \leq 3dD\sqrt{2\beta\gamma} + \frac{6D\sqrt{2\beta\gamma}}{1+\alpha}T^{(1+\alpha)/2} + 3\beta dD^2 + 3D(G + \beta D)d\ln T,$$

$$A + C \leq 3dGD + \frac{4G(G + \beta D)d(1 + \ln T)}{\beta} + 4dG\sqrt{\frac{2\gamma}{\beta}} + \sqrt{\frac{2\gamma}{\beta}}\frac{8G}{1+\alpha}T^{(1+\alpha)/2},$$

where $A = \sum_{t=1}^{T} G\|\mathbf{y}_{\tau_t} - \mathbf{y}_{\tau_{t'}}\|_2$, $C = \sum_{t=s}^{T+d-1} \sum_{i=\tau_t}^{\tau_{t+1}-1} G\|\mathbf{y}_{\tau_t} - \mathbf{y}_i\|_2$, and $E = \sum_{t=1}^{T} \frac{3\beta D}{2}\|\mathbf{y}_t - \mathbf{y}_{\tau_t}\|_2$.

Note that Lemma 4 also introduces an assumption about $\mathbf{y}_t$ and $F_{t-1}(\mathbf{y})$. According to our Algorithm 2, $\mathbf{y}_t$ is actually generated by approximately minimizing $F_{t-1}(\mathbf{y})$ with a linear optimization step. Therefore, by following the analysis of OFW for strongly convex losses [Wan and Zhang, 2021], we show that this assumption is satisfied with $\gamma = 16(G + 2\beta D)^2/\beta$ and $\alpha = 1/3$.

**Lemma 5** *Let $\mathbf{y}_t^* = \operatorname{argmin}_{\mathbf{y} \in \mathcal{K}} F_{t-1}(\mathbf{y})$ for any $t = 2, \ldots, T+1$, where $F_t(\mathbf{y})$ is defined in (12). Suppose Assumption 1 and 2 hold, and all losses are $\beta$-strongly convex. For any $t = 2, \ldots, T+1$, Algorithm 2 has*

$$F_{t-1}(\mathbf{y}_t) - F_{t-1}(\mathbf{y}_t^*) \leq \frac{16(G + 2\beta D)^2 (t-1)^{1/3}}{\beta}.$$

By combining Lemmas 4 and 5, we have

$$A + C + E = O\left(T^{(1+\alpha)/2} + d\log T\right) = O(T^{2/3} + d\log T). \tag{20}$$

Furthermore, by following the analysis of OFW for strongly convex losses [Wan and Zhang, 2021], we establish an upper bound for the third term in (19).

**Lemma 6** *Suppose Assumption 1 and 2 hold, and all losses are $\beta$-strongly convex. For any $\mathbf{x}^* \in \mathcal{K}$, Algorithm 2 ensures*

$$B' \leq \frac{6\sqrt{2}(G + 2\beta D)^2 T^{2/3}}{\beta} + \frac{2(G + 2\beta D)^2 \ln T}{\beta} + (G + \beta D)D$$

*where $B' = \sum_{t=1}^{T} \langle \mathbf{g}_{c_t}, \mathbf{y}_t - \mathbf{x}^* \rangle - \sum_{t=1}^{T} \frac{\beta}{2}\|\mathbf{y}_t - \mathbf{x}^*\|_2^2$.*

Finally, by combining (19), (20), and Lemma 6, we complete this proof.

## 5 Experiments

In this section, we conduct simulation experiments to verify the performance of our delayed OFW for convex losses and strongly convex losses. All algorithms are implemented wtih Matlab R2017a and tested on a laptop with 2.4GHz CPU and 16GB memory. The code is available from `https://github.com/yuanyuwan/NeurIPS22`.

Specifically, we consider a delayed OCO problem with $T = 10000$ total rounds over the decision set $\mathcal{K} = \left\{ \mathbf{x} \in \mathbb{R}^{1000} \big| \|\mathbf{x}\|_1 \leq 200 \right\}$, which satisfies Assumption 2 with $D = 400$ due to $\|\mathbf{x} - \mathbf{y}\|_2 \leq \|\mathbf{x}\|_2 + \|\mathbf{y}\|_2 \leq \|\mathbf{x}\|_1 + \|\mathbf{y}\|_1 \leq 400$ for any $\mathbf{x}, \mathbf{y} \in \mathcal{K}$. Inspired by Li et al. [2019], in each round $t \in [T]$, the adversary chooses the following loss function

$$f_t(\mathbf{x}) = \|\mathbf{x}\|_2 + \langle \mathbf{b}_t, \mathbf{x} \rangle$$

where each element of $\mathbf{b}_t \in \mathbb{R}^{1000}$ is independently and uniformly sampled from $[-1, 1]$. For any $\mathbf{x} \in \mathcal{K}$, since $\nabla f_t(\mathbf{x}) = 2\mathbf{x} + \mathbf{b}_t$, it is easy to verify that $\|\nabla f_t(\mathbf{x})\|_2 \leq 2\|\mathbf{x}\|_2 + \|\mathbf{b}_t\|_2 \leq 2\|\mathbf{x}\|_1 + \|\mathbf{b}_t\|_2 \leq 400 + \sqrt{1000}$, which implies that this loss function satisfies Assumption 1 with $G = 400 + \sqrt{1000}$. We also notice that this loss function satisfies the definition of $\beta$-strongly convex function (i.e., Definition 1) with $\beta = 2$. Moreover, different values of the maximum delay $d$ in the set $\{1, 51, 101, \ldots, 501\}$ are tried in our experiments. For each specific $d$, to simulate arbitrary delays, $d_t$ is independently and uniformly sampled from the set $\{\lceil 0.1d \rceil, \lceil 0.1d \rceil + 1, \lceil 0.1d \rceil + 2, \ldots, d\}$.

We compare our proposed algorithms against the combination of the technique in Joulani et al. [2013] and OFW. Our delayed OFW for convex losses and strongly convex losses are denoted as DOFW (i.e., Algorithm 1) and DOFW$_{\text{sc}}$ (i.e., Algorithm 2), respectively. Similarly, since the technique in Joulani et al. [2013] is named as black-box online learning under delayed feedback (BOLD), we denote the combination of BOLD with OFW for convex losses and strongly convex losses as BOLD-OFW and BOLD-OFW$_{\text{sc}}$, respectively. The parameters of all algorithms are set as what their corresponding theories suggest. Specifically, according to our Theorems 1 and 2, we set $\eta = \frac{D}{\sqrt{2}G(T+2)^{3/4}}$ for our DOFW and set $\beta = 2$ for our DOFW$_{\text{sc}}$. According to Joulani et al. [2013], BOLD-OFW

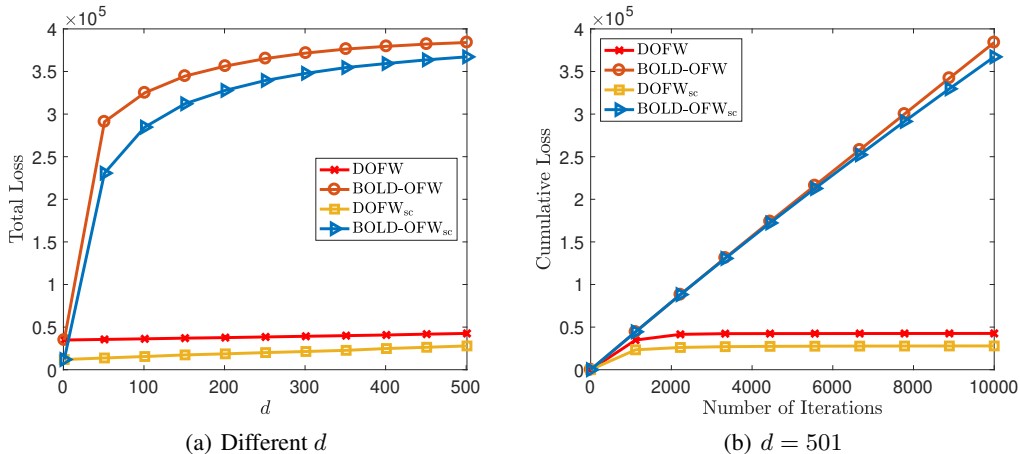

(a) Different $d$                     (b) $d = 501$

Figure 1: Comparisons of our DOFW and DOFW$_{\text{sc}}$ against BOLD-OFW and BOLD-OFW$_{\text{sc}}$.

and BOLD-OFW$_{\text{sc}}$ will maintain several instances of OFW for convex and strongly convex losses, respectively. Note that in the non-delayed case with $d = 1$ our delayed OFW actually reduces to the original OFW. Therefore, our Theorems 1 and 2 can also be utilized to choose the parameters for each instance of OFW maintained in BOLD-OFW and BOLD-OFW$_{\text{sc}}$. To be precise, in BOLD-OFW, we set $\eta = \frac{D}{\sqrt{2}G(T/d+2)^{3/4}}$ for each instance of OFW for convex losses, since the total rounds of each instance is roughly $T/d$. In BOLD-OFW$_{\text{sc}}$, we only need to set $\beta = 2$ for each instance of OFW for strongly convex losses. Moreover, for our delayed OFW and each instance of OFW, the initial decision is set to $\mathbf{1}/50$, where $\mathbf{1}$ denotes the all-ones vector.

Fig. 1(a) shows the total loss of $T$ rounds for each algorithm under different values of the maximum delay $d$. First, when $d = 1$, the total loss of our DOFW is the same as that of BOLD-OFW and the total loss of our DOFW$_{\text{sc}}$ is the same as that of BOLD-OFW$_{\text{sc}}$, which is reasonable because in this case DOFW and BOLD-OFW reduce to the original OFW for convex losses, and DOFW$_{\text{sc}}$ and BOLD-OFW$_{\text{sc}}$ reduce to the original OFW for strongly convex losses. Second, for $d = 51, 101, 151, \ldots, 501$, our DOFW and DOFW$_{\text{sc}}$ are better than BOLD-OFW and BOLD-OFW$_{\text{sc}}$ respectively, which clearly verifies the advantage of our algorithms in the delayed setting. It is worthy to notice that $d = 501$ is larger than $T^{2/3}$. Moreover, for our DOFW and DOFW$_{\text{sc}}$, when $d$ increases from 1 to 501, the growth of the total loss is very slow, which is consistent with the dependence of our regret bounds on $d$. Fig. 1(b) shows the cumulative loss for each algorithm when $d = 501$. As the number of iterations increases, the cumulative loss of BOLD-OFW and BOLD-OFW$_{\text{sc}}$ increase much faster than that of our algorithms.

## 6 Conclusion and future work

In this paper, we propose delayed OFW for OCO with arbitrary delays. For convex losses, we show that it attains an $O(T^{3/4} + dT^{1/4})$ regret bound, which matches the $O(T^{3/4})$ regret bound of OFW in the non-delayed setting, as long as $d$ does not exceed $O(\sqrt{T})$. When losses are strongly convex, we further prove that it can attain an $O(T^{2/3} + d\log T)$ regret bound, which matches the $O(T^{2/3})$ regret bound of OFW in the non-delayed setting, as long as $d$ does not exceed $O(T^{2/3}/\log T)$. Simulation experiments demonstrate the performance of delayed OFW in the delayed setting.

This paper only extends the classical OFW to the delayed setting. In the future, we will investigate how to develop delayed variants for other projection-free online algorithms.

## Acknowledgments

This work was partially supported by NSFC (61921006, 62122037), and JiangsuSF (BK20200064).

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
