# Supplementary Material

## A Proof of Lemma 1

We first note that $F_t(\mathbf{y})$ is 2-strongly convex for any $t = 0, \ldots, T$, and Hazan and Kale [2012] have proved that for any $\beta$-strongly convex function $f(\mathbf{x})$ over $\mathcal{K}$ and any $\mathbf{x} \in \mathcal{K}$, it holds that

$$\frac{\beta}{2}\|\mathbf{x} - \mathbf{x}^*\|_2^2 \le f(\mathbf{x}) - f(\mathbf{x}^*) \tag{21}$$

where $\mathbf{x}^* = \operatorname{argmin}_{\mathbf{x} \in \mathcal{K}} f(\mathbf{x})$.

Then, we consider the term $A = \sum_{t=1}^{T} G\|\mathbf{y}_{\tau_t} - \mathbf{y}_{\tau_{t'}}\|_2$. If $T \le 2d$, we have

$$A = \sum_{t=1}^{T} G\|\mathbf{y}_{\tau_t} - \mathbf{y}_{\tau_{t'}}\|_2 \le TGD \le 2dGD \tag{22}$$

where the first inequality is due to Assumption 2. If $T > 2d$, we have

$$A = \sum_{t=1}^{2d} G\|\mathbf{y}_{\tau_t} - \mathbf{y}_{\tau_{t'}}\|_2 + \sum_{t=2d+1}^{T} G\|\mathbf{y}_{\tau_t} - \mathbf{y}_{\tau_{t'}}\|_2$$

$$\le 2dGD + \sum_{t=2d+1}^{T} G(\|\mathbf{y}_{\tau_t} - \mathbf{y}_{\tau_t}^*\|_2 + \|\mathbf{y}_{\tau_t}^* - \mathbf{y}_{\tau_{t'}}^*\|_2 + \|\mathbf{y}_{\tau_{t'}}^* - \mathbf{y}_{\tau_{t'}}\|_2). \tag{23}$$

Because of (21), for any $t \in [T+1]$, we have

$$\|\mathbf{y}_t - \mathbf{y}_t^*\|_2 \le \sqrt{F_{t-1}(\mathbf{y}_t) - F_{t-1}(\mathbf{y}_t^*)} \le \sqrt{\gamma}(t+2)^{-\alpha/2} \tag{24}$$

where the last inequality is due to $F_{t-1}(\mathbf{y}_t) - F_{t-1}(\mathbf{y}_t^*) \le \gamma(t+2)^{-\alpha}$.

Moreover, for any $i \ge \tau_t$, we have

$$\|\mathbf{y}_{\tau_t}^* - \mathbf{y}_i^*\|_2^2 \le F_{i-1}(\mathbf{y}_{\tau_t}^*) - F_{i-1}(\mathbf{y}_i^*)$$

$$= F_{\tau_t - 1}(\mathbf{y}_{\tau_t}^*) - F_{\tau_t - 1}(\mathbf{y}_i^*) + \left\langle \eta \sum_{k=\tau_t}^{i-1} \mathbf{g}_{c_k}, \mathbf{y}_{\tau_t}^* - \mathbf{y}_i^* \right\rangle$$

$$\le \eta \left\| \sum_{k=\tau_t}^{i-1} \mathbf{g}_{c_k} \right\|_2 \|\mathbf{y}_{\tau_t}^* - \mathbf{y}_i^*\|_2$$

$$\le \eta G(i - \tau_t)\|\mathbf{y}_{\tau_t}^* - \mathbf{y}_i^*\|_2 \tag{25}$$

where the first inequality is still due to (21) and the last inequality is due to Assumption 1.

Because of $t' = t + d_t - 1 \ge t$, we have $\tau_{t'} \ge \tau_t$. Then, from (25), we have

$$\|\mathbf{y}_{\tau_t}^* - \mathbf{y}_{\tau_{t'}}^*\|_2 \le \eta G(\tau_{t'} - \tau_t) = \eta G \sum_{k=t}^{t'-1} |\mathcal{F}_k|. \tag{26}$$

Then, by substituting (24) and (26) into (23), if $T > 2d$, we have

$$A \le 2dGD + \sum_{t=2d+1}^{T} G\left(\sqrt{\gamma}(\tau_t + 2)^{-\alpha/2} + \eta G \sum_{k=t}^{t'-1} |\mathcal{F}_k| + \sqrt{\gamma}(\tau_{t'} + 2)^{-\alpha/2}\right)$$

$$\le 2dGD + \sum_{t=2d+1}^{T} 2G\sqrt{\gamma}(\tau_t + 2)^{-\alpha/2} + \eta G^2 \sum_{t=2d+1}^{T} \sum_{k=t}^{t'-1} |\mathcal{F}_k| \tag{27}$$

$$\le 2dGD + \sum_{t=2d+1}^{T} 2G\sqrt{\gamma}(\tau_t - 1)^{-\alpha/2} + \eta G^2 \sum_{t=2d+1}^{T} \sum_{k=t}^{t'-1} |\mathcal{F}_k|$$

where the second inequality is due to $(\tau_t + 2)^{-\alpha/2} \ge (\tau_{t'} + 2)^{-\alpha/2}$ for $\tau_t \le \tau_{t'}$ and $\alpha > 0$.

To bound the second term in the right side of (27), we introduce the following lemma.

**Lemma 7** *Let $\tau_t = 1 + \sum_{i=1}^{t-1} |\mathcal{F}_i|$ for any $t \in [T+d]$. If $T > 2d$, for $0 < \alpha \leq 1$, we have*

$$\sum_{t=2d+1}^{T} (\tau_t - 1)^{-\alpha/2} \leq d + \frac{2}{2-\alpha} T^{1-\alpha/2}. \tag{28}$$

For the third term in the right side of (27), if $T > 2d$, we have

$$\sum_{t=2d+1}^{T} \sum_{k=t}^{t'-1} |\mathcal{F}_k| \leq \sum_{t=1}^{T} \sum_{k=t}^{t'-1} |\mathcal{F}_k| \leq \sum_{t=1}^{T} \sum_{k=t}^{t+d-1} |\mathcal{F}_k| = \sum_{k=0}^{d-1} \sum_{t=1+k}^{T+k} |\mathcal{F}_t|$$

$$\leq \sum_{k=0}^{d-1} \sum_{t=1}^{T+d-1} |\mathcal{F}_t| = dT \tag{29}$$

where the second inequality is due to

$$t' - 1 < t' = t + d_t - 1 \leq t + d - 1.$$

By substituting (28) and (29) into (27) and combining with (22), we have

$$A \leq 2dGD + 2Gd\sqrt{\gamma} + \frac{4G\sqrt{\gamma}}{2-\alpha} T^{1-\alpha/2} + \eta G^2 dT. \tag{30}$$

Then, for the term $C = \sum_{t=s}^{T+d-1} \sum_{i=\tau_t}^{\tau_{t+1}-1} G\|\mathbf{y}_{\tau_t} - \mathbf{y}_i\|_2$, we have

$$C = \sum_{i=\tau_s}^{\tau_{s+1}-1} G\|\mathbf{y}_{\tau_t} - \mathbf{y}_i\|_2 + \sum_{t=s+1}^{T+d-1} \sum_{i=\tau_t}^{\tau_{t+1}-1} G\|\mathbf{y}_{\tau_t} - \mathbf{y}_i\|_2$$

$$\leq |\mathcal{F}_s|GD + \sum_{t=s+1}^{T+d-1} \sum_{i=\tau_t}^{\tau_{t+1}-1} G(\|\mathbf{y}_{\tau_t} - \mathbf{y}_{\tau_t}^*\|_2 + \|\mathbf{y}_{\tau_t}^* - \mathbf{y}_i^*\|_2 + \|\mathbf{y}_i^* - \mathbf{y}_i\|_2)$$

$$\leq |\mathcal{F}_s|GD + \sum_{t=s+1}^{T+d-1} \sum_{i=\tau_t}^{\tau_{t+1}-1} G\left(\sqrt{\gamma}(\tau_t + 2)^{-\alpha/2} + \eta G(i - \tau_t) + \sqrt{\gamma}(i+2)^{-\alpha/2}\right) \tag{31}$$

$$\leq |\mathcal{F}_s|GD + \sum_{t=s+1}^{T+d-1} \sum_{i=\tau_t}^{\tau_{t+1}-1} 2G\sqrt{\gamma}(\tau_t + 2)^{-\alpha/2} + \eta G^2 \sum_{t=s+1}^{T+d-1} \sum_{k=0}^{\tau_{t+1}-\tau_t-1} k$$

$$\leq |\mathcal{F}_s|GD + \sum_{t=s+1}^{T+d-1} \sum_{i=\tau_t}^{\tau_{t+1}-1} 2G\sqrt{\gamma}(\tau_t - 1)^{-\alpha/2} + \eta G^2 \sum_{t=s}^{T+d-1} \sum_{k=0}^{\tau_{t+1}-\tau_t-1} k$$

where the first inequality is due to Assumption 2, the second inequality is due to (24) and (25), and the third inequality is due to $(\tau_t + 2)^{-\alpha/2} \geq (i+2)^{-\alpha/2}$ for $\tau_t \leq i$ and $\alpha > 0$.

Moreover, for any $t \in [T + d - 1]$ and $k \in \mathcal{F}_t$, since $1 \leq d_k \leq d$, we have

$$t - d + 1 \leq k = t - d_k + 1 \leq t$$

which implies that

$$|\mathcal{F}_t| \leq t - (t - d + 1) + 1 = d. \tag{32}$$

Then, it is easy to verify that

$$\tau_{t+1} - \tau_t - 1 < \tau_{t+1} - \tau_t = |\mathcal{F}_t| \leq d.$$

Therefore, by combining with (31), we have

$$C \leq dGD + \sum_{t=s+1}^{T+d-1} \sum_{i=\tau_t}^{\tau_{t+1}-1} 2G\sqrt{\gamma}(\tau_t - 1)^{-\alpha/2} + \eta G^2 \sum_{t=s}^{T+d-1} \frac{|\mathcal{F}_t|^2}{2}$$

$$\leq dGD + \sum_{t=s+1}^{T+d-1} \sum_{i=\tau_t}^{\tau_{t+1}-1} 2G\sqrt{\gamma}(\tau_t - 1)^{-\alpha/2} + \eta G^2 \sum_{t=s}^{T+d-1} \frac{d|\mathcal{F}_t|}{2} \tag{33}$$

$$= dGD + \sum_{t=s+1}^{T+d-1} \sum_{i=\tau_t}^{\tau_{t+1}-1} 2G\sqrt{\gamma}(\tau_t - 1)^{-\alpha/2} + \frac{\eta G^2 dT}{2}.$$

Furthermore, we introduce the following lemma.

**Lemma 8** *Let $\tau_t = 1 + \sum_{i=1}^{t-1} |\mathcal{F}_i|$ for any $t \in [T+d]$ and $s = \min \{t | t \in [T+d-1], |\mathcal{F}_t| > 0\}$. For $0 < \alpha \le 1$, we have*

$$\sum_{t=s+1}^{T+d-1} \sum_{i=\tau_t}^{\tau_{t+1}-1} (\tau_t - 1)^{-\alpha/2} \le d + \frac{2}{2-\alpha} T^{1-\alpha/2}. \tag{34}$$

By substituting (34) into (33), we have

$$C \le dGD + 2G\sqrt{\gamma}d + \frac{4G\sqrt{\gamma}}{2-\alpha} T^{1-\alpha/2} + \frac{\eta G^2 dT}{2} \tag{35}$$

We complete the proof by combing (30) and (35).

## B  Proof of Lemma 2

At the beginning of this proof, we recall the standard definition for smooth functions [Boyd and Vandenberghe, 2004].

**Definition 2** *A function $f(\mathbf{x}) : \mathcal{K} \to \mathbb{R}$ is called $\alpha$-smooth over $\mathcal{K}$ if for all $\mathbf{x}, \mathbf{y} \in \mathcal{K}$, it holds that $f(\mathbf{y}) \le f(\mathbf{x}) + \langle \nabla f(\mathbf{x}), \mathbf{y} - \mathbf{x} \rangle + \frac{\alpha}{2} \|\mathbf{y} - \mathbf{x}\|_2^2$.*

It is not hard to verify that $F_t(\mathbf{y})$ is 2-smooth over $\mathcal{K}$ for any $t \in [T]$. This property will be utilized in the following.

For brevity, we define $h_t = F_{t-1}(\mathbf{y}_t) - F_{t-1}(\mathbf{y}_t^*)$ for $t = 1, \ldots, T+1$ and $h_t(\mathbf{y}_{t-1}) = F_{t-1}(\mathbf{y}_{t-1}) - F_{t-1}(\mathbf{y}_t^*)$ for $t = 2, \ldots, T+1$.

For $t = 1$, since $\mathbf{y}_1 = \operatorname{argmin}_{\mathbf{y} \in \mathcal{K}} \|\mathbf{y} - \mathbf{y}_1\|_2^2$, we have

$$h_1 = F_0(\mathbf{y}_1) - F_0(\mathbf{y}_1^*) = 0 \le \frac{8D^2}{\sqrt{3}} = \frac{8D^2}{\sqrt{t+2}}. \tag{36}$$

Then, for any $T + 1 \ge t \ge 2$, we have

$$\begin{aligned}
h_t(\mathbf{y}_{t-1}) &= F_{t-1}(\mathbf{y}_{t-1}) - F_{t-1}(\mathbf{y}_t^*) \\
&= F_{t-2}(\mathbf{y}_{t-1}) - F_{t-2}(\mathbf{y}_t^*) + \langle \eta \mathbf{g}_{c_{t-1}}, \mathbf{y}_{t-1} - \mathbf{y}_t^* \rangle \\
&\le F_{t-2}(\mathbf{y}_{t-1}) - F_{t-2}(\mathbf{y}_{t-1}^*) + \langle \eta \mathbf{g}_{c_{t-1}}, \mathbf{y}_{t-1} - \mathbf{y}_t^* \rangle \\
&\le h_{t-1} + \eta \|\mathbf{g}_{c_{t-1}}\|_2 \|\mathbf{y}_{t-1} - \mathbf{y}_t^*\|_2 \\
&\le h_{t-1} + \eta \|\mathbf{g}_{c_{t-1}}\|_2 \|\mathbf{y}_{t-1} - \mathbf{y}_{t-1}^*\|_2 + \eta \|\mathbf{g}_{c_{t-1}}\|_2 \|\mathbf{y}_{t-1}^* - \mathbf{y}_t^*\|_2 \\
&\le h_{t-1} + \eta G \|\mathbf{y}_{t-1} - \mathbf{y}_{t-1}^*\|_2 + \eta G \|\mathbf{y}_{t-1}^* - \mathbf{y}_t^*\|_2
\end{aligned} \tag{37}$$

where the first inequality is due to $\mathbf{y}_{t-1}^* = \operatorname{argmin}_{\mathbf{y} \in \mathcal{K}} F_{t-2}(\mathbf{y})$ and the last inequality is due to Assumption 1.

Moreover, for any $T + 1 \ge t \ge 2$, we note that $F_{t-2}(\mathbf{x})$ is also 2-strongly convex, which implies that

$$\|\mathbf{y}_{t-1} - \mathbf{y}_{t-1}^*\|_2 \le \sqrt{F_{t-2}(\mathbf{y}_{t-1}) - F_{t-2}(\mathbf{y}_{t-1}^*)} \le \sqrt{h_{t-1}} \tag{38}$$

where the first inequality is due to (21).

Similarly, for any $T + 1 \ge t \ge 2$

$$\begin{aligned}
\|\mathbf{y}_{t-1}^* - \mathbf{y}_t^*\|_2^2 &\le F_{t-1}(\mathbf{y}_{t-1}^*) - F_{t-1}(\mathbf{y}_t^*) \\
&= F_{t-2}(\mathbf{y}_{t-1}^*) - F_{t-2}(\mathbf{y}_t^*) + \langle \eta \mathbf{g}_{c_{t-1}}, \mathbf{y}_{t-1}^* - \mathbf{y}_t^* \rangle \\
&\le \eta \|\mathbf{g}_{c_{t-1}}\|_2 \|\mathbf{y}_{t-1}^* - \mathbf{y}_t^*\|_2
\end{aligned}$$

which implies that

$$\|\mathbf{y}_{t-1}^* - \mathbf{y}_t^*\|_2 \le \eta \|\mathbf{g}_{c_{t-1}}\|_2 \le \eta G. \tag{39}$$

By combining (37), (38), and (39), for any $T + 1 \ge t \ge 2$, we have

$$h_t(\mathbf{y}_{t-1}) \le h_{t-1} + \eta G \sqrt{h_{t-1}} + \eta^2 G^2. \tag{40}$$

Then, for any $T + 1 \geq t \geq 2$, since $F_{t-1}(\mathbf{y})$ is 2-smooth, we have

$$
\begin{aligned}
h_t &= F_{t-1}(\mathbf{y}_t) - F_{t-1}(\mathbf{y}_t^*) \\
&= F_{t-1}(\mathbf{y}_{t-1} + \sigma_{t-1}(\mathbf{v}_{t-1} - \mathbf{y}_{t-1})) - F_{t-1}(\mathbf{y}_t^*) \\
&\leq h_t(\mathbf{y}_{t-1}) + \langle \nabla F_{t-1}(\mathbf{y}_{t-1}), \sigma_{t-1}(\mathbf{v}_{t-1} - \mathbf{y}_{t-1}) \rangle + \sigma_{t-1}^2 \|\mathbf{v}_{t-1} - \mathbf{y}_{t-1}\|_2^2.
\end{aligned}
\tag{41}
$$

Moreover, for any $t \in [T]$, according to Algorithm 1, we have

$$
\sigma_t = \underset{\sigma \in [0,1]}{\operatorname{argmin}} \langle \sigma(\mathbf{v}_t - \mathbf{y}_t), \nabla F_t(\mathbf{y}_t) \rangle + \sigma^2 \|\mathbf{v}_t - \mathbf{y}_t\|_2^2.
\tag{42}
$$

Therefore, for $t = 2$, by combining (40) and (41), we have

$$
\begin{aligned}
h_2 &\leq h_1 + \eta G \sqrt{h_1} + \eta^2 G^2 + \langle \nabla F_1(\mathbf{y}_1), \sigma_1(\mathbf{v}_1 - \mathbf{y}_1) \rangle + \sigma_1^2 \|\mathbf{v}_1 - \mathbf{y}_1\|_2^2 \\
&\leq h_1 + \eta G \sqrt{h_1} + \eta^2 G^2 = \frac{D^2}{2(T+2)^{3/2}} \leq 4D^2 = \frac{8D^2}{\sqrt{t+2}}
\end{aligned}
\tag{43}
$$

where the second inequality is due to (42), and the first equality is due to (36) and $\eta = \frac{D}{\sqrt{2}G(T+2)^{3/4}}$.

Then, for any $t = 3, \ldots, T+1$, by defining $\sigma'_{t-1} = 2/\sqrt{t+1}$ and assuming $h_{t-1} \leq \frac{8D^2}{\sqrt{t+1}}$, we have

$$
\begin{aligned}
h_t &\leq h_t(\mathbf{y}_{t-1}) + \langle \nabla F_{t-1}(\mathbf{y}_{t-1}), \sigma'_{t-1}(\mathbf{v}_{t-1} - \mathbf{y}_{t-1}) \rangle + (\sigma'_{t-1})^2 \|\mathbf{v}_{t-1} - \mathbf{y}_{t-1}\|_2^2 \\
&\leq h_t(\mathbf{y}_{t-1}) + \langle \nabla F_{t-1}(\mathbf{y}_{t-1}), \sigma'_{t-1}(\mathbf{y}_t^* - \mathbf{y}_{t-1}) \rangle + (\sigma'_{t-1})^2 \|\mathbf{v}_{t-1} - \mathbf{y}_{t-1}\|_2^2 \\
&\leq (1 - \sigma'_{t-1}) h_t(\mathbf{y}_{t-1}) + (\sigma'_{t-1})^2 \|\mathbf{v}_{t-1} - \mathbf{y}_{t-1}\|_2^2 \\
&\leq (1 - \sigma'_{t-1})(h_{t-1} + \eta G \sqrt{h_{t-1}} + \eta^2 G^2) + (\sigma'_{t-1})^2 D^2 \\
&\leq (1 - \sigma'_{t-1}) h_{t-1} + \eta G \sqrt{h_{t-1}} + \eta^2 G^2 + (\sigma'_{t-1})^2 D^2 \\
&\leq \left(1 - \frac{2}{\sqrt{t+1}}\right) \frac{8D^2}{\sqrt{t+1}} + \frac{2D^2}{(T+2)^{3/4}(t+1)^{1/4}} + \frac{D^2}{2(T+2)^{3/2}} + \frac{4D^2}{t+1} \\
&\leq \left(1 - \frac{2}{\sqrt{t+1}}\right) \frac{8D^2}{\sqrt{t+1}} + \frac{2D^2}{t+1} + \frac{D^2}{2(t+1)} + \frac{4D^2}{t+1} \\
&\leq \left(1 - \frac{2}{\sqrt{t+1}}\right) \frac{8D^2}{\sqrt{t+1}} + \frac{8D^2}{t+1} \\
&= \left(1 - \frac{1}{\sqrt{t+1}}\right) \frac{8D^2}{\sqrt{t+1}} \leq \frac{8D^2}{\sqrt{t+2}}
\end{aligned}
\tag{44}
$$

where the first inequality is due to (41) and (42), the second inequality is due to $\mathbf{v}_{t-1} \in \operatorname{argmin}_{\mathbf{y} \in \mathcal{K}} \langle \nabla F_{t-1}(\mathbf{y}_{t-1}), \mathbf{y} \rangle$, the third inequality is due to the convexity of $F_{t-1}(\mathbf{y})$, the fourth inequality is due to (40), and the last inequality is due to

$$
\left(1 - \frac{1}{\sqrt{t+1}}\right) \frac{1}{\sqrt{t+1}} \leq \frac{1}{\sqrt{t+2}}
\tag{45}
$$

for any $t \geq 0$.

Note that (45) can be derived by dividing $(t+1)\sqrt{t+2}$ into both sides of the following inequality

$$
\sqrt{t+2}\sqrt{t+1} - \sqrt{t+2} \leq (\sqrt{t+1}+1)\sqrt{t+1} - \sqrt{t+2} \leq t+1+\sqrt{t+1} - \sqrt{t+2} \leq t+1.
$$

By combining (36), (43), and (44), we complete this proof.

## C   Proof of Lemma 3

In the beginning, we define $\mathbf{y}_t^* = \operatorname{argmin}_{\mathbf{y} \in \mathcal{K}} F_{t-1}(\mathbf{y})$ for any $t \in [T+1]$, where $F_t(\mathbf{y}) = \eta \sum_{i=1}^t \langle \mathbf{g}_{c_i}, \mathbf{y} \rangle + \|\mathbf{y} - \mathbf{y}_1\|_2^2$.

Then, it is easy to verify that

$$
\sum_{t=1}^T \langle \mathbf{g}_{c_t}, \mathbf{y}_t - \mathbf{x}^* \rangle = \sum_{t=1}^T \langle \mathbf{g}_{c_t}, \mathbf{y}_t - \mathbf{y}_t^* \rangle + \sum_{t=1}^T \langle \mathbf{g}_{c_t}, \mathbf{y}_t^* - \mathbf{x}^* \rangle.
\tag{46}
$$

Therefore, we will continue to upper bound the right side of (46). By applying Lemma 2, we have

$$\sum_{t=1}^{T} \langle \mathbf{g}_{c_t}, \mathbf{y}_t - \mathbf{y}_t^* \rangle \leq \sum_{t=1}^{T} \|\mathbf{g}_{c_t}\|_2 \|\mathbf{y}_t - \mathbf{y}_t^*\|_2 \leq \sum_{t=1}^{T} G \sqrt{F_{t-1}(\mathbf{y}_t) - F_{t-1}(\mathbf{y}_t^*)}$$
$$\leq \sum_{t=1}^{T} \frac{2\sqrt{2}GD}{(t+2)^{1/4}} \leq \frac{8\sqrt{2}GD(T+2)^{3/4}}{3} \tag{47}$$

where the second inequality is due to (21) and Assumption 1, and the last inequality is due to $\sum_{t=1}^{T}(t+2)^{-1/4} \leq 4(T+2)^{3/4}/3$.

Then, to bound $\sum_{t=1}^{T} \langle \mathbf{g}_{c_t}, \mathbf{y}_t^* - \mathbf{x}^* \rangle$, we introduce the following lemma.

**Lemma 9** *(Lemma 6.6 of Garber and Hazan [2016]) Let $\{f_t(\mathbf{y})\}_{t=1}^{T}$ be a sequence of loss functions and let $\mathbf{y}_t^* \in \operatorname{argmin}_{\mathbf{y} \in \mathcal{K}} \sum_{i=1}^{t} f_i(\mathbf{y})$ for any $t \in [T]$. Then, it holds that*

$$\sum_{t=1}^{T} f_t(\mathbf{y}_t^*) - \min_{\mathbf{y} \in \mathcal{K}} \sum_{t=1}^{T} f_t(\mathbf{y}) \leq 0.$$

To apply Lemma 9, we define $\tilde{f}_1(\mathbf{y}) = \eta \langle \mathbf{g}_{c_1}, \mathbf{y} \rangle + \|\mathbf{y} - \mathbf{y}_1\|_2^2$ and $\tilde{f}_t(\mathbf{y}) = \eta \langle \mathbf{g}_{c_t}, \mathbf{y} \rangle$ for any $t \geq 2$. Note that $F_t(\mathbf{y}) = \sum_{i=1}^{t} \tilde{f}_i(\mathbf{y})$ and $\mathbf{y}_{t+1}^* = \operatorname{argmin}_{\mathbf{y} \in \mathcal{K}} F_t(\mathbf{y})$ for any $t = 1, \ldots, T$. Then, by applying Lemma 9 to $\{\tilde{f}_t(\mathbf{y})\}_{t=1}^{T}$, we have

$$\sum_{t=1}^{T} \tilde{f}_t(\mathbf{y}_{t+1}^*) - \sum_{t=1}^{T} \tilde{f}_t(\mathbf{x}^*) \leq 0$$

which implies that

$$\eta \sum_{t=1}^{T} \langle \mathbf{g}_{c_t}, \mathbf{y}_{t+1}^* - \mathbf{x}^* \rangle \leq \|\mathbf{x}^* - \mathbf{y}_1\|_2^2 - \|\mathbf{y}_2^* - \mathbf{y}_1\|_2^2.$$

According to Assumption 2, we have

$$\sum_{t=1}^{T} \langle \mathbf{g}_{c_t}, \mathbf{y}_{t+1}^* - \mathbf{x}^* \rangle \leq \frac{1}{\eta} \|\mathbf{x}^* - \mathbf{y}_1\|_2^2 \leq \frac{D^2}{\eta}.$$

Then, we have

$$\sum_{t=1}^{T} \langle \mathbf{g}_{c_t}, \mathbf{y}_t^* - \mathbf{x}^* \rangle = \sum_{t=1}^{T} \langle \mathbf{g}_{c_t}, \mathbf{y}_{t+1}^* - \mathbf{x}^* \rangle + \sum_{t=1}^{T} \langle \mathbf{g}_{c_t}, \mathbf{y}_t^* - \mathbf{y}_{t+1}^* \rangle$$
$$\leq \frac{D^2}{\eta} + \sum_{t=1}^{T} \|\mathbf{g}_{c_t}\|_2 \|\mathbf{y}_t^* - \mathbf{y}_{t+1}^*\|_2 \tag{48}$$
$$\leq \frac{D^2}{\eta} + \eta T G^2$$
$$\leq \sqrt{2}GD(T+2)^{3/4} + \frac{GDT^{1/4}}{\sqrt{2}}$$

where the second inequality is due to (39) and Assumption 1, and the last inequality is due to $\eta = \frac{D}{\sqrt{2}G(T+2)^{3/4}}$.

By substituting (47) and (48) into (46), we complete the proof.

## D    Proof of Lemma 4

We first consider the term $E = \sum_{t=1}^{T} \frac{3\beta D}{2} \|\mathbf{y}_t - \mathbf{y}_{\tau_t}\|_2$. If $T \leq 2d$, it is easy to verify that

$$E = \sum_{t=1}^{T} \frac{3\beta D}{2} \|\mathbf{y}_t - \mathbf{y}_{\tau_t}\|_2 \leq \frac{3\beta T D^2}{2} \leq 3\beta d D^2 \tag{49}$$

where the first inequality is due to Assumption 2.

Then, if $T > 2d$, we have

$$E = \frac{3\beta D}{2} \sum_{t=1}^{2d} \|\mathbf{y}_t - \mathbf{y}_{\tau_t}\|_2 + \frac{3\beta D}{2} \sum_{t=2d+1}^{T} \|\mathbf{y}_t - \mathbf{y}_{\tau_t}\|_2$$

$$\leq 3\beta d D^2 + \frac{3\beta D}{2} \sum_{t=2d+1}^{T} \left( \|\mathbf{y}_t - \mathbf{y}_t^*\|_2 + \|\mathbf{y}_t^* - \mathbf{y}_{\tau_t}^*\|_2 + \|\mathbf{y}_{\tau_t}^* - \mathbf{y}_{\tau_t}\|_2 \right). \tag{50}$$

Because $F_{t-1}(\mathbf{y})$ is $(t-1)\beta$-strongly convex for any $t = 2, \ldots, T+1$, we have

$$\|\mathbf{y}_t - \mathbf{y}_t^*\|_2 \leq \sqrt{\frac{2(F_{t-1}(\mathbf{y}_t) - F_{t-1}(\mathbf{y}_t^*))}{(t-1)\beta}} \leq \sqrt{\frac{2\gamma}{(t-1)^{1-\alpha}\beta}} \tag{51}$$

where the first inequality is due to (21) and the second inequality is due to $F_{t-1}(\mathbf{y}_t) - F_{t-1}(\mathbf{y}_t^*) \leq \gamma(t-1)^\alpha$.

Before considering $\|\mathbf{y}_t^* - \mathbf{y}_{\tau_t}^*\|_2$, we define $\tilde{f}_t(\mathbf{y}) = \langle \mathbf{g}_{c_t}, \mathbf{y} \rangle + \frac{\beta}{2}\|\mathbf{y} - \mathbf{y}_t\|_2^2$ for any $t = 1, \ldots, T$. Note that $F_t(\mathbf{y}) = \sum_{i=1}^{t} \tilde{f}_i(\mathbf{y})$. Moreover, for any $\mathbf{x}, \mathbf{y} \in \mathcal{K}$ and $t = 1, \ldots, T$, we have

$$|\tilde{f}_t(\mathbf{x}) - \tilde{f}_t(\mathbf{y})| = \left| \langle \mathbf{g}_{c_t}, \mathbf{x} - \mathbf{y} \rangle + \frac{\beta}{2}\|\mathbf{x} - \mathbf{y}_t\|_2^2 - \frac{\beta}{2}\|\mathbf{y} - \mathbf{y}_t\|_2^2 \right|$$

$$= \left| \langle \mathbf{g}_{c_t}, \mathbf{x} - \mathbf{y} \rangle + \frac{\beta}{2}\langle \mathbf{x} - \mathbf{y}_t + \mathbf{y} - \mathbf{y}_t, \mathbf{x} - \mathbf{y} \rangle \right| \tag{52}$$

$$\leq \|\mathbf{g}_{c_t}\|_2 \|\mathbf{x} - \mathbf{y}\|_2 + \frac{\beta}{2}(\|\mathbf{x} - \mathbf{y}_t\|_2 + \|\mathbf{y} - \mathbf{y}_t\|_2)\|\mathbf{x} - \mathbf{y}\|_2$$

$$\leq (G + \beta D)\|\mathbf{x} - \mathbf{y}\|_2$$

where the last inequality is due to Assumptions 1 and 2.

Because of (21), for any $i \geq j > 1$, we have

$$\|\mathbf{y}_j^* - \mathbf{y}_i^*\|_2^2 \leq \frac{2(F_{i-1}(\mathbf{y}_j^*) - F_{i-1}(\mathbf{y}_i^*))}{(i-1)\beta}$$

$$= \frac{2(F_{j-1}(\mathbf{y}_j^*) - F_{j-1}(\mathbf{y}_i^*)) + 2\sum_{k=j}^{i-1}\left(\tilde{f}_k(\mathbf{y}_j^*) - \tilde{f}_k(\mathbf{y}_i^*)\right)}{(i-1)\beta} \tag{53}$$

$$\leq \frac{2(i-j)(G + \beta D)\|\mathbf{y}_j^* - \mathbf{y}_i^*\|_2}{(i-1)\beta}$$

where the last inequality is due to $\mathbf{y}_j^* = \operatorname{argmin}_{\mathbf{y} \in \mathcal{K}} F_{j-1}(\mathbf{y})$ and (52).

Note that all gradients queried at rounds $1, \ldots, t-d$ must arrive before round $t$. Therefore, for any $t \geq 2d+1$, we have $\tau_t = 1 + \sum_{k=1}^{t-1}|\mathcal{F}_k| \geq t - d + 1 > t - d$ and

$$\|\mathbf{y}_t^* - \mathbf{y}_{\tau_t}^*\|_2 \leq \frac{2(t - \tau_t)(G + \beta D)}{(t-1)\beta} \leq \frac{2d(G + \beta D)}{(t-1)\beta} \tag{54}$$

where the first inequality is due to $t \geq \tau_t > 1$ and (53).

By combining (50) with (51) and (54), if $T > 2d$, we have

$$E \leq 3\beta d D^2 + \frac{3\beta D}{2} \sum_{t=2d+1}^{T} \left( \sqrt{\frac{2\gamma}{(t-1)^{1-\alpha}\beta}} + \frac{2d(G + \beta D)}{(t-1)\beta} + \sqrt{\frac{2\gamma}{(\tau_t-1)^{1-\alpha}\beta}} \right)$$

$$\leq 3\beta d D^2 + 3\beta D \sum_{t=2d+1}^{T} \sqrt{\frac{2\gamma}{(\tau_t-1)^{1-\alpha}\beta}} + 3D(G + \beta D)d\sum_{t=2}^{T}\frac{1}{t}$$

$$\leq 3\beta d D^2 + 3\beta D \sum_{t=2d+1}^{T} \sqrt{\frac{2\gamma}{(\tau_t-1)^{1-\alpha}\beta}} + 3D(G + \beta D)d\ln T$$

$$\leq 3\beta d D^2 + 3dD\sqrt{2\beta\gamma} + \frac{6D\sqrt{2\beta\gamma}}{1+\alpha}T^{(1+\alpha)/2} + 3D(G + \beta D)d\ln T$$

where the second inequality is due to $(\tau_t - 1)^{1-\alpha} \le (t-1)^{1-\alpha}$ for $t \ge \tau_t > 1$ and $\alpha < 1$, and the last inequality is due to Lemma 7 and $0 < 1 - \alpha \le 1$.

By combining (49) with the above inequality, we have

$$E \le 3\beta dD^2 + 3dD\sqrt{2\beta\gamma} + \frac{6D\sqrt{2\beta\gamma}}{1+\alpha}T^{(1+\alpha)/2} + 3D(G+\beta D)d\ln T.$$

Then, we proceed to bound the term $C = \sum_{t=s}^{T+d-1}\sum_{i=\tau_t}^{\tau_{t+1}-1}G\|\mathbf{y}_{\tau_t} - \mathbf{y}_i\|_2$. Similar to (31), we first have

$$C \le |\mathcal{F}_s|GD + \sum_{t=s+1}^{T+d-1}\sum_{i=\tau_t}^{\tau_{t+1}-1}G(\|\mathbf{y}_{\tau_t} - \mathbf{y}_{\tau_t}^*\|_2 + \|\mathbf{y}_{\tau_t}^* - \mathbf{y}_i^*\|_2 + \|\mathbf{y}_i^* - \mathbf{y}_i\|_2). \tag{55}$$

By combining (55) with $|\mathcal{F}_s| \le d$, (51), and (53), we have

$$
\begin{aligned}
C \le{}& dGD + \sum_{t=s+1}^{T+d-1}\sum_{i=\tau_t}^{\tau_{t+1}-1}G\left(\sqrt{\frac{2\gamma}{(\tau_t-1)^{1-\alpha}\beta}} + \frac{2(i-\tau_t)(G+\beta D)}{(i-1)\beta} + \sqrt{\frac{2\gamma}{(i-1)^{1-\alpha}\beta}}\right)\\
\le{}& dGD + \sum_{t=s+1}^{T+d-1}\sum_{i=\tau_t}^{\tau_{t+1}-1}G\left(2\sqrt{\frac{2\gamma}{(\tau_t-1)^{1-\alpha}\beta}} + \frac{2(i-\tau_t)(G+\beta D)}{(i-1)\beta}\right)\\
\le{}& dGD + 2dG\sqrt{\frac{2\gamma}{\beta}} + \sqrt{\frac{2\gamma}{\beta}}\frac{4G}{1+\alpha}T^{(1+\alpha)/2} + \sum_{t=s+1}^{T+d-1}\sum_{i=\tau_t}^{\tau_{t+1}-1}\frac{2dG(G+\beta D)}{(i-1)\beta}
\end{aligned}
\tag{56}
$$

where the first inequality is due to $(\tau_t - 1)^{1-\alpha} \le (i-1)^{1-\alpha}$ for $0 < \tau_t - 1 \le i - 1$ and $\alpha < 1$, and the last inequality is due to Lemma 8, $0 < 1 - \alpha \le 1$, and $i - \tau_t \le \tau_{t+1} - 1 - \tau_t \le |\mathcal{F}_t| \le d$.

Recall that we have defined

$$\mathcal{I}_t = \begin{cases} \emptyset, & \text{if } |\mathcal{F}_t| = 0,\\ \{\tau_t, \tau_t + 1, \ldots, \tau_{t+1} - 1\}, & \text{otherwise.} \end{cases}$$

It is not hard to verify that

$$\cup_{t=s+1}^{T+d-1}\mathcal{I}_t = \{|F_s| + 1, \ldots, T\}, \mathcal{I}_i \cap \mathcal{I}_j = \emptyset, \forall i \ne j. \tag{57}$$

By combining (57) with (56), we have

$$
\begin{aligned}
C \le{}& dGD + 2dG\sqrt{\frac{2\gamma}{\beta}} + \sqrt{\frac{2\gamma}{\beta}}\frac{4G}{1+\alpha}T^{(1+\alpha)/2} + \sum_{t=|F_s|+1}^{T}\frac{2dG(G+\beta D)}{(t-1)\beta}\\
\le{}& dGD + 2dG\sqrt{\frac{2\gamma}{\beta}} + \sqrt{\frac{2\gamma}{\beta}}\frac{4G}{1+\alpha}T^{(1+\alpha)/2} + \sum_{t=2}^{T}\frac{2dG(G+\beta D)}{(t-1)\beta}\\
\le{}& dGD + 2dG\sqrt{\frac{2\gamma}{\beta}} + \sqrt{\frac{2\gamma}{\beta}}\frac{4G}{1+\alpha}T^{(1+\alpha)/2} + \frac{2dG(G+\beta D)(1+\ln T)}{\beta}.
\end{aligned}
\tag{58}
$$

Next, we proceed to bound the term $A = \sum_{t=1}^{T}G\|\mathbf{y}_{\tau_t} - \mathbf{y}_{\tau_{t'}}\|_2$. Similar to (23), if $T > 2d$, we have

$$
\begin{aligned}
A \le{}& 2dGD + \sum_{t=2d+1}^{T}G(\|\mathbf{y}_{\tau_t} - \mathbf{y}_{\tau_t}^*\|_2 + \|\mathbf{y}_{\tau_t}^* - \mathbf{y}_{\tau_{t'}}^*\|_2 + \|\mathbf{y}_{\tau_{t'}}^* - \mathbf{y}_{\tau_{t'}}\|_2)\\
\le{}& 2dGD + \sum_{t=2d+1}^{T}G\left(\sqrt{\frac{2\gamma}{(\tau_t-1)^{1-\alpha}\beta}} + \frac{2(\tau_{t'}-\tau_t)(G+\beta D)}{(\tau_{t'}-1)\beta} + \sqrt{\frac{2\gamma}{(\tau_{t'}-1)^{1-\alpha}\beta}}\right)\\
\le{}& 2dGD + \sum_{t=2d+1}^{T}2G\sqrt{\frac{2\gamma}{(\tau_t-1)^{1-\alpha}\beta}} + \sum_{t=2d+1}^{T}\frac{2G(G+\beta D)}{\beta}\sum_{k=t}^{t'-1}\frac{|\mathcal{F}_k|}{\sum_{i=1}^{k}|\mathcal{F}_i|}
\end{aligned}
\tag{59}
$$

where the second inequality is due to (51) and (53), and the last inequality is due to $\tau_{t'} \ge \tau_t > 1$ and

$$\frac{(\tau_{t'} - \tau_t)}{(\tau_{t'} - 1)} = \frac{\sum_{k=t}^{t'-1}|\mathcal{F}_k|}{\sum_{k=1}^{t'-1}|\mathcal{F}_k|} \le \sum_{k=t}^{t'-1}\frac{|\mathcal{F}_k|}{\sum_{i=1}^{k}|\mathcal{F}_i|}.$$

Then, we introduce the following lemma.

**Lemma 10** *Let $h_k = \sum_{i=1}^{k} |\mathcal{F}_i|$. If $T > 2d$, we have*

$$\sum_{t=2d+1}^{T} \sum_{k=t}^{t'-1} \frac{|\mathcal{F}_k|}{h_k} \leq d + d\ln T.$$

By applying Lemmas 7 and 10 to (59) and combining with (22), we have

$$A \leq 2dGD + 2dG\sqrt{\frac{2\gamma}{\beta}} + \sqrt{\frac{2\gamma}{\beta}} \frac{4G}{1+\alpha} T^{(1+\alpha)/2} + \frac{2G(G + \beta D)d(1 + \ln T)}{\beta}. \tag{60}$$

Finally, by combining (58) and (60), we complete this proof.

# E  Proof of Lemmas 5 and 6

Recall that $F_\tau(\mathbf{y})$ defined in Algorithm 2 is equivalent to that defined in (12). Let $\tilde{f}_t(\mathbf{y}) = \langle \mathbf{g}_{c_t}, \mathbf{y} \rangle + \frac{\beta}{2}\|\mathbf{y} - \mathbf{y}_t\|_2^2$ for any $t = 1, \ldots, T$, which is $\beta$-strongly convex. Moreover, as proved in (52), functions $\tilde{f}_1(\mathbf{y}), \ldots, \tilde{f}_T(\mathbf{y})$ are $(G + \beta D)$-Lipschitz over $\mathcal{K}$ (see the definition of Lipschitz functions in Hazan [2016]). Then, because of $\nabla \tilde{f}_t(\mathbf{y}_t) = \mathbf{g}_{c_t}$, it is not hard to verify that decisions $\mathbf{y}_1, \ldots, \mathbf{y}_{T+1}$ in our Algorithm 2 are actually generated by performing OFW for strongly convex losses (see Algorithm 2 in Wan and Zhang [2021] for details) on functions $\tilde{f}_1(\mathbf{y}), \ldots, \tilde{f}_T(\mathbf{y})$. Note that when Assumption 2 holds, and functions $\tilde{f}_1(\mathbf{y}), \ldots, \tilde{f}_T(\mathbf{y})$ are $\beta$-strongly convex and $G'$-Lipschitz, Lemma 6 of Wan and Zhang [2021] has already shown that

$$F_{t-1}(\mathbf{y}_t) - F_{t-1}(\mathbf{y}_t^*) \leq \frac{16(G' + \beta D)^2(t - 1)^{1/3}}{\beta}$$

for any $t = 2, \ldots, T+1$. Therefore, our Lemma 5 can be derived by simply substituting $G' = G + \beta D$ into the above inequality.

Moreover, when Assumption 2 holds, and functions $\tilde{f}_1(\mathbf{y}), \ldots, \tilde{f}_T(\mathbf{y})$ are $\beta$-strongly convex and $G'$-Lipschitz, Theorem 3 of Wan and Zhang [2021] has already shown that

$$\sum_{t=1}^{T} \tilde{f}_t(\mathbf{y}_t) - \sum_{t=1}^{T} \tilde{f}_t(\mathbf{x}^*) \leq \frac{6\sqrt{2}(G' + \beta D)^2 T^{2/3}}{\beta} + \frac{2(G' + \beta D)^2 \ln T}{\beta} + G'D.$$

We notice that $\sum_{t=1}^{T} \left( \langle \mathbf{g}_{c_t}, \mathbf{y}_t - \mathbf{x}^* \rangle - \frac{\beta}{2}\|\mathbf{y}_t - \mathbf{x}^*\|_2^2 \right) = \sum_{t=1}^{T} \tilde{f}_t(\mathbf{y}_t) - \sum_{t=1}^{T} \tilde{f}_t(\mathbf{x}^*)$. Therefore, our Lemma 6 can be derived by simply substituting $G' = G + \beta D$ into the above inequality.

# F  Proof of Lemma 7

Since the gradient $\mathbf{g}_1$ must arrive before round $d + 1$, for any $T \geq t \geq 2d + 1$, it is easy to verify that $\tau_t = 1 + \sum_{i=1}^{t-1} |\mathcal{F}_i| \geq 1 + \sum_{i=1}^{d+1} |\mathcal{F}_i| \geq 2$. Moreover, for any $i \geq 2$ and $(i+1)d \geq t \geq id + 1$, since all gradients queried at rounds $1, \ldots, (i-1)d + 1$ must arrive before round $id + 1$, we have

$$\tau_t = 1 + \sum_{i=1}^{t-1} |\mathcal{F}_i| \geq (i-1)d + 2. \tag{61}$$

Then, we have

$$\sum_{t=2d+1}^{T} (\tau_t - 1)^{-\alpha/2} = \sum_{t=2d+1}^{\lfloor T/d \rfloor d} (\tau_t - 1)^{-\alpha/2} + \sum_{t=\lfloor T/d \rfloor d + 1}^{T} (\tau_t - 1)^{-\alpha/2}$$

$$\leq \sum_{i=2}^{\lfloor T/d \rfloor - 1} \sum_{t=id+1}^{(i+1)d} (\tau_t - 1)^{-\alpha/2} + d \leq d + \sum_{i=2}^{\lfloor T/d \rfloor - 1} d((i-1)d + 1)^{-\alpha/2}$$

$$\leq d + \sum_{i=2}^{\lfloor T/d \rfloor - 1} d^{1-\alpha/2}(i-1)^{-\alpha/2} \leq d + \sum_{i=1}^{\lfloor T/d \rfloor} d^{1-\alpha/2} i^{-\alpha/2}$$

$$\leq d + \frac{2}{2-\alpha} d^{1-\alpha/2} \left( \lfloor T/d \rfloor \right)^{1-\alpha/2} \leq d + \frac{2}{2-\alpha} T^{1-\alpha/2}$$

where the first inequality is due to $(\tau_t - 1)^{-\alpha/2} \leq 1$ for $\alpha > 0$ and $\tau_t \geq 2$, and the second inequality is due to (61) and $\alpha > 0$.

## G  Proof of Lemma 8

Because of $\tau_t = 1 + \sum_{i=1}^{t-1} |\mathcal{F}_i|$, we have

$$
\begin{aligned}
&\sum_{t=s+1}^{T+d-1} \sum_{i=\tau_t}^{\tau_{t+1}-1} (\tau_t - 1)^{-\alpha/2} \\
&= \sum_{t=s+1}^{T+d-1} \frac{|\mathcal{F}_t|}{(\sum_{i=s}^{t-1} |\mathcal{F}_i|)^{\alpha/2}} \\
&= \sum_{t=s+1}^{T+d-1} \frac{|\mathcal{F}_t|}{(\sum_{i=s}^{t} |\mathcal{F}_i|)^{\alpha/2}} + \sum_{t=s+1}^{T+d-1} |\mathcal{F}_t| \left( \frac{1}{(\sum_{i=s}^{t-1} |\mathcal{F}_i|)^{\alpha/2}} - \frac{1}{(\sum_{i=s}^{t} |\mathcal{F}_i|)^{\alpha/2}} \right) \quad (62) \\
&\leq \sum_{t=s+1}^{T+d-1} \frac{|\mathcal{F}_t|}{(\sum_{i=s}^{t} |\mathcal{F}_i|)^{\alpha/2}} + \sum_{t=s+1}^{T+d-1} d \left( \frac{1}{(\sum_{i=s}^{t-1} |\mathcal{F}_i|)^{\alpha/2}} - \frac{1}{(\sum_{i=s}^{t} |\mathcal{F}_i|)^{\alpha/2}} \right) \\
&\leq \sum_{t=s+1}^{T+d-1} \frac{|\mathcal{F}_t|}{(\sum_{i=s}^{t} |\mathcal{F}_i|)^{\alpha/2}} + \frac{d}{|\mathcal{F}_s|^{\alpha/2}} \leq \sum_{t=s+1}^{T+d-1} \frac{|\mathcal{F}_t|}{(\sum_{i=s}^{t} |\mathcal{F}_i|)^{\alpha/2}} + d
\end{aligned}
$$

where the first inequality is due to (32) and $(\sum_{i=s}^{t-1} |\mathcal{F}_i|)^{\alpha/2} \leq (\sum_{i=s}^{t} |\mathcal{F}_i|)^{\alpha/2}$.

Let $h_t = \sum_{i=s}^{t} |\mathcal{F}_i|$ for any $t = s, \ldots, T+d-1$. Since $0 < \alpha \leq 1$, it is not hard to verify that

$$
\begin{aligned}
\sum_{t=s+1}^{T+d-1} \frac{|\mathcal{F}_t|}{(\sum_{i=s}^{t} |\mathcal{F}_i|)^{\alpha/2}} &= \sum_{t=s+1}^{T+d-1} \frac{|\mathcal{F}_t|}{(h_t)^{\alpha/2}} = \sum_{t=s+1}^{T+d-1} \int_{h_{t-1}}^{h_t} \frac{1}{(h_t)^{\alpha/2}} dx \\
&\leq \sum_{t=s+1}^{T+d-1} \int_{h_{t-1}}^{h_t} \frac{1}{x^{\alpha/2}} dx = \int_{h_s}^{h_{T+d-1}} \frac{1}{x^{\alpha/2}} dx = \int_{|\mathcal{F}_s|}^{T} \frac{1}{x^{\alpha/2}} dx \quad (63) \\
&\leq \frac{2}{2-\alpha} T^{1-\alpha/2}.
\end{aligned}
$$

Finally, we complete this proof by combining (62) with (63).

## H  Proof of Lemma 10

It is not hard to verify that

$$
\begin{aligned}
\sum_{t=2d+1}^{T} \sum_{k=t}^{t'-1} \frac{|\mathcal{F}_k|}{h_k} &\leq \sum_{t=s}^{T} \sum_{k=t}^{t'-1} \frac{|\mathcal{F}_k|}{h_k} \leq \sum_{t=s}^{T} \sum_{k=t}^{t+d-1} \frac{|\mathcal{F}_k|}{h_k} = \sum_{k=0}^{d-1} \sum_{t=s+k}^{T+k} \frac{|\mathcal{F}_t|}{h_t} \\
&\leq \sum_{k=0}^{d-1} \sum_{t=s}^{T+d-1} \frac{|\mathcal{F}_t|}{h_t} = d \sum_{t=s}^{T+d-1} \frac{|\mathcal{F}_t|}{h_t}
\end{aligned}
$$

where the first inequality is due to $s \leq d < 2d+1$, and the second inequality is due to $t' - 1 = t + d_t - 2 < t + d - 1$.

Moreover, we have

$$
\begin{aligned}
\sum_{t=s}^{T+d-1} \frac{|\mathcal{F}_t|}{h_t} &= \frac{|\mathcal{F}_s|}{h_s} + \sum_{t=s+1}^{T+d-1} \int_{h_{t-1}}^{h_t} \frac{1}{h_t} dx \leq \frac{|\mathcal{F}_s|}{h_s} + \sum_{t=s+1}^{T+d-1} \int_{h_{t-1}}^{h_t} \frac{1}{x} dx \\
&= \frac{|\mathcal{F}_s|}{h_s} + \int_{h_s}^{h_{T+d-1}} \frac{1}{x} dx = 1 + \ln \frac{T}{|\mathcal{F}_s|} \leq 1 + \ln T
\end{aligned}
$$

where the last equality is due to $h_s = |\mathcal{F}_s|$ and $h_{T+d-1} = T$.

Finally, we complete this proof by combining the above two inequalities.