# OpenReview forum: "Online Frank-Wolfe with Arbitrary Delays"
_NeurIPS.cc/2022/Conference — NeurIPS 2022 Accept_

### Official Review · Reviewer_6sxN · 2022-07-08

**Rating:** 4
**Confidence:** 5
**Soundness:** 2 fair
**Presentation:** 3 good
**Contribution:** 2 fair

**Summary:**

This paper mainly proposed a delayed variant of OFW called Online Frank-Wolfe with Arbitrary Delays, which allows gradients to be delayed by arbitrary rounds. The algorithm can avoid the limitation of the original OFW, as well as solving the problem of Online convex optimization (OCO) that needs to compute a projection onto the decision set to ensure the feasibility of each decision. By performing an update similar to OFW after receiving any delayed gradient, and playing the latest decision for each round, this algorithm could enjoy the same regret bound like that of the original OFW.

**Questions:**

 This paper only theoretically analyzes the effect of the proposed algorithm, and does not carry out more specific experiments to qualitatively and quantitatively verify the actual effect.

in addition, Elad Hazan et al.  in the manuscript titled"  Faster Projection-free Online Learning" give an efficient projection-free algorithm that guarantees O(T2/3) regret for general online convex optimization with smooth cost functions and one linear optimization computation per iteration.
Please discuss the relationship with these previous research results, and how much has it improved.




**Ethics Review Area:**

["I don’t know"]

**Limitations:**

I don't  know the limitations and potential negative societal impact of their work.

**Strengths And Weaknesses:**

Strength:

（1）This paper extends the classical OFW to the delayed setting, which requires the fewer storage and computational resources compared with the non-delayed setting.
（2）Unlike other algorithms, i.e., Weinberger and Ordentlich proposed a technique that can convert any traditional OCO algorithm for the non-delayed setting into this delayed setting, the Online Frank-Wolfe with Arbitrary Delays provided in this paper can achieve better regret bound for convex losses and strongly convex losses.
（3）The delayed variant of OFW enjoys the same regret bound like that of the original OFW, which has a certain positive impact on the convex optimization field, and has higher practical value than the original OFW.
（4）This paper only theoretically analyzes the effect of the proposed algorithm, and does not carry out more specific experiments to verify the actual effect.

Weaknesses:

（1）This paper only theoretically analyzes the effect of the proposed algorithm, and does not carry out more specific experiments to verify the actual effect. It is necessary to do some experiments.
（2）This paper only extends the classical OFW to the delayed setting, there still are more delayed variants for other projection-free online algorithms. It is meaningful to do more investigations.

---

> ### Author Response · Authors · 2022-08-01
> **Response to Reviewer 6sxN**
>
> Many thanks for the constructive reviews! We believe there are some misunderstandings, and have clarified them below. We hope the reviewer could reevaluate our paper, and are very happy to respond more questions during the reviewer-author discussion period.
>
> ---
>
> Q1: It is necessary to do some experiments.
>
> A1: Thanks for the suggestion. Please check our common response to all reviewers, which introduces our experimental results completed during the rebuttal period.
>
> ---
>
> Q2: This paper only extends the classical OFW to the delayed setting, there still are more delayed variants for other projection-free online algorithms. It is meaningful to do more investigations.
>
> A2: Thanks for the suggestion. We agree that it is meaningful to investigate other projection-free online algorithms in the delayed setting, which was actually listed as a future work in our Section 6.
>
> However, other projection-free algorithms could be very different from OFW in both the algorithm design and theoretical analysis. So, it is somehow unreasonable to investigate them in one paper.
>
> Moreover, we would like to emphasize that the main reason for us to consider OFW is because it is the best-known projection-free online algorithm for convex and strongly convex losses without additional assumptions.
>
> ---
>
> Q3: In addition, Elad Hazan et al. in the manuscript titled" Faster Projection-free Online Learning" give an efficient projection-free algorithm that guarantees $O(T^{2/3})$ regret for general online convex optimization with smooth cost functions and one linear optimization computation per iteration. Please discuss the relationship with these previous research results, and how much has it improved.
>
> A3: Thanks for the suggestion. However, we would like to clarify that the method of Hazan and Minasyan [2020] (i.e., the work you pointed out) was already introduced and discussed in our manuscript.
>
> Specifically, from lines 83 to 85 of our manuscript, we have introduced their method and its regret bound. From lines 100 to 101, we have emphasized that these previous studies about projection-free online algorithms, including Hazan and Minasyan [2020] did not consider the practical problem of delayed feedback.
>
> In summary, Hazan and Minasyan [2020] propose a projection-free method for *convex and smooth functions in the non-delayed setting*, and our work investigates the classical OFW method for *convex functions in the delayed setting*.

---

### Official Review · Reviewer_LySe · 2022-07-11

**Rating:** 6
**Confidence:** 3
**Soundness:** 2 fair
**Presentation:** 3 good
**Contribution:** 2 fair

**Summary:**

This work studies the problem of online convex optimization where the gradient of the function at the query point is revealed after an arbitrary delay. The authors proposed a delayed version of the online Frank-Wolfe algorithm, for convex (Algorithm 1, page 5) and strongly convex (Algorithm 2, page 6). In the case when the maximum of the delays is sufficiently small, the proposed method can recover the regret bounds for convex and strongly convex loss functions.

**Questions:**

1. Can one use similar ideas for providing a delayed version of online mirror descent algorithm?

**Limitations:**

The authors addressed the limitations of their work.

**Strengths And Weaknesses:**

Strengths:

The problem of delayed feedback is well motivated. The paper is very well written and easy to understand. The main contribution of the paper is in the case when the maximum of the delays is sufficiently small, the proposed method can recover the regret bounds for convex and strongly convex losses (outlined in lines 64 - 69)4. The authors provide a complete literature review. They clearly discussed their contributions and improvement to the prior work.

Weaknesses:
1. It would be nice if the authors provide a numerical comparison with Joulani et al. [2013].

---

> ### Author Response · Authors · 2022-08-01
> **Response to Reviewer LySe**
>
> Q1: It would be nice if the authors provide a numerical comparison with Joulani et al. [2013].
>
> A1: Thanks for the suggestion. Please check our common response to all reviewers, which introduces our experimental results completed during the rebuttal period.
>
> ---
>
> Q2: Can one use similar ideas for providing a delayed version of online mirror descent algorithm?
>
> A2: One can use similar ideas to develop a delayed version of online mirror descent (OMD). However, as explained in our response to Q2 of Reviewer cKYY, our delayed OFW can keep the same regret bound as OFW mainly due to the additive effect between the linear optimization step and the delay. Since OMD does not have a similar property, for the delayed version of OMD, it is hard to keep the same regret bound as OMD.

---

> > ### Comment · Reviewer_LySe · 2022-08-06
> > **Acknowledging and further suggestions**
> >
> > I would like to thank the authors for their complete response. The authors addressed my concerns.
> >
> > In point 2, the authors mentioned that "...our delayed OFW can keep the same regret bound as OFW mainly due to the additive effect between the linear optimization step and the delay." I think it is very important to mention this in the main text (along with A2 to Reviewer cKYY), and I believe that this is the main reason that the results of the paper are limited to OFW. I would encourage the authors to provide a discussion on this in the revision.

---

> > > ### Author Response · Authors · 2022-08-07
> > > **Response to the suggestion**
> > >
> > > Thanks for the suggestion. We will revise our paper accordingly.

---

### Official Review · Reviewer_rmw7 · 2022-07-11

**Rating:** 6
**Confidence:** 3
**Soundness:** 3 good
**Presentation:** 3 good
**Contribution:** 3 good

**Summary:**

This paper studies online convex optimization with delayed gradients. The authors study the Online Frank-Wolfe (OFW) algorithm, which simply runs the FW update for all the gradients as soon as they arrive. The authors prove regret bounds in the convex and strongly convex settings, which still match the original regret bound for FW, as long as the delay does not exceed certain threshold.

**Questions:**

Is it clear whether OFW is the optimal algorithm for delayed feedback?

**Limitations:**

Yes

**Strengths And Weaknesses:**

Interesting problem with simple algorithm and nice results. The paper is well-written and the analysis seems clear. The resulting regret bounds are robust to delay up to some threshold, which is surprising and nice.

---

> ### Author Response · Authors · 2022-08-01
> **Response to Reviewer rmw7**
>
> Q1: Is it clear whether OFW is the optimal algorithm for delayed feedback?
>
> A1: If we only consider the regret, our delayed OFW is not the best algorithm for the delayed setting. One can simply combine the technique in Joulani et al. [2013] with the classical OGD method to achieve $O(\sqrt{dT})$ and $O(d \log T)$ regret bounds for convex and strongly convex losses respectively, which are better than our regret bounds.
>
> However, it is worthy to emphasize that our delayed OFW belongs to projection-free algorithms, which are much more efficient for complex decision sets than those projection-based algorithms such as OGD. To the best of our knowledge, among projection-free online algorithms, the regret bounds of our delayed OFW for convex and strongly convex losses are optimal when the term involving $d$ is not dominant.

---

### Official Review · Reviewer_cKYY · 2022-07-12

**Rating:** 7
**Confidence:** 2
**Soundness:** 3 good
**Presentation:** 3 good
**Contribution:** 3 good

**Summary:**

In this paper, the authors consider the problem of online convex optimisation where at each time $t\leq T$ the player chooses an action $x_t \in \mathcal{K}$ and receives a payment $f_t(x_t)$, and where the gradient $\nabla f_t(x_t)$ is only revealed after an arbitrary delay $d_t$ smaller than some constant $d$. The aim of the player is to minimize its regret.

The authors propose a modified version of the online Frank-Wolfe algorithm that can adapt to the delays. Under classical assumptions on the boundedness of the gradient and on the diameter of $\mathcal{K}$, the authors bound the regret of their algorithm in the convex case and in the strongly convex case. They show that in the convex case, the regret is $O(T^{3/4} + dT^{1/4})$, while in the strongly convex case, it is $O(T^{2/3} + d \log(T))$. These regret rates improve over previous rates for online convex optimisation with delays. Moreover, when $d$ is small enough, these rates match the rates of the online Frank-Wolf algorithm without delay.

**Questions:**

Is the OWF algorithm particularly suited for delayed feedback because of the form of $F_t$ (that takes into account all previously observed gradients)? Or do you think you could achieve the same rates with other classical algorithm for online convex optimisation? Could you maybe provide a little more intuition on why delay do not lead to an increase in the reget?

**Limitations:**

The authors have properly addressed the limitation of the theoretical aspects of their work, however one could have expected their paper to also include a comparison of the empirical performances of their algorithm with benchmark methods.

**Strengths And Weaknesses:**

Originality : The authors modify the Frank-Wolfe algorithm to take into account delays in the observation of the gradients. The provide a theoretical analysis of the regret of this algorithm.

Quality : This paper is technically solid. The lack of simulations comparing their algorithms to benchmark algorithms is somewhat disappointing.

Clarity : This paper is clearly written. The assumptions are justified. The motivations and the literature review well addressed.

Significance : This paper presents a clear contribution by adapting a classical algorithm to a well motivated, practical problem.

---

> ### Author Response · Authors · 2022-08-01
> **Response to Reviewer cKYY**
>
> Q1: one could have expected their paper to also include a comparison of the empirical performances of their algorithm with benchmark methods.
>
> A1: Thanks for the suggestion. Please check our common response to all reviewers, which introduces our experimental results completed during the rebuttal period.
>
> ---
>
> Q2: Could you maybe provide a little more intuition on why delay do not lead to an increase in the regret?
>
> A2: To help understanding, we first notice that in the non-delayed setting, the regret bound of OFW is worse than that of OGD, because the linear optimization (LO) step utilized in OFW brings an additional error than the projection operation utilized in OGD.
>
> In the delayed setting, it is natural to conjecture that both the LO step and the delay will affect the regret of our delayed OFW. Fortunately, due to the following two reasons, our delayed OFW attains the same regret bound as OFW for a relatively large amount of delay.
> 1. The combined effect of the LO step and the delay can be proved to be additive, instead of being multiplicative.
> 2. The effect of the delay is weak than that of the LO step for a relatively large amount of delay, which does not change the order of the regret bound.
>
> ---
>
> Q3: Is the OWF algorithm particularly suited for delayed feedback because of the form of $F_t$ (that takes into account all previously observed gradients)? Or do you think you could achieve the same rates with other classical algorithm for online convex optimization?
>
> A3: First, as explained in A2, our delayed OFW can keep the same regret bound as OFW mainly due to the additive effect between the linear optimization (LO) step and the delay, instead of the form of $F_t$.
>
> Moreover, as explained below, the classical regularized follow the leader (RFTL) algorithm [Hazan, 2016] actually uses observed gradients in a way similar to OFW, but we cannot keep its regret bound unchanged in the delayed setting.
> 1. Similar to OFW, RFTL also uses all previously observed gradients to define a surrogate loss function $F_t(\mathbf{x})$ (Eq. (2) in our manuscript). However, different from the LO step in OFW (Eq. (1) in our manuscript), RFTL updates the decision to the exact minimizer of $F_t(\mathbf{x})$.
> 2. In the non-delayed setting, RFTL can attain a regret bound of $O(\eta T+\eta^{-1})$, where $\eta$ is a parameter in $F_t(\mathbf{x})$. By setting $\eta=O(1/\sqrt{T})$, we can get a regret bound of $O(\sqrt{T})$.
> 3. RFTL can be extended into the delayed setting in a way similar to our delayed OFW. However, by following the proof of our Theorem 1 with slight modifications, we can only achieve a regret bound of $O(\eta T+\eta^{-1}+\eta dT)$. Then, the best achievable regret bound is $O(\sqrt{dT})$, which is worse than $O(\sqrt{T})$ in the non-delayed setting.
>
> The above example also indicates that it is hard to extend other algorithms to the delayed setting while keeping the regret bound the same as that in the non-delayed setting.

---

> > ### Comment · Reviewer_cKYY · 2022-08-04
> > **Response to the authors**
> >
> > I thank the authors for their detailed response, which answered all my questions.

---

### Author Response · Authors · 2022-08-01
**Common Response to All Reviewers**

We thank all the reviewers for your detailed comments. In the following, we first respond to the common suggestion about experiments, and other questions are addressed in a separate response for every reviewer. Please let us know if you have any further questions.

During the rebuttal period, we have revised our manuscript by adding experimental results, which are presented in a new section named Experiments. Specifically, we conduct simulation experiments to compare our delayed OFW against the combination of the technique in Joulani et al. [2013] and OFW. A delayed OCO problem with strongly convex losses over $T=10000$ rounds is considered, and different values of the maximum delay $d$ in the set {$1, 51, 101, …, 501$} are tried. It is worthy to notice that $d=501$ is larger than $T^{2/3}$.

Let DOFW and DOFW$_\mathrm{sc}$ denote our delayed OFW for convex losses and strongly convex losses. Similarly, let BOLD-OFW and BOLD-OFW$_\mathrm{sc}$ denote the combination of the technique in Joulani et al. [2013] with OFW for convex losses and strongly convex losses. From Figure 1 in our revised manuscript, we have the following main observations.
1. For $d=51, 101, 151, …, 501$, our DOFW and DOFW$_\mathrm{sc}$ are better than BOLD-OFW and BOLD-OFW$_\mathrm{sc}$ respectively, which clearly verifies the advantage of our algorithms in the delayed setting.
2. For our DOFW and DOFW$_\mathrm{sc}$, when $d$ increases from $1$ to $501$, the growth of the total loss is very slow, which is consistent with the dependence of our regret bounds on $d$.

---

### Meta-Review · Area_Chair_z6Qz · 2022-08-24

**Recommendation:** Accept
**Confidence:** Certain

**Metareview:**

Following a discussion with the authors, all reviewers were in favor of accepting except for Reviewer 6sxN whose main concern is around the experimental evaluation.  From my own look into the paper, I tend to agree with the reviewers: the paper is well-written, it solves a natural (even if a bit niche) problem with a simple algorithm; and the theoretical analysis is well-executed and even somewhat surprising: the effect of the delay turns out to be additive, rather than multiplicative as is the case in other online optimization settings.  The concerns around experimental evaluations are valid, but I do not feel that strong experiments are crucial in such a theory-focused paper.

All considered, I gladly recommend to accept the paper.

**Award:**

No

---

### Decision · Program_Chairs · 2022-09-14

Accept